

# Estimating total alkalinity for coastal ocean acidification monitoring at regional to continental scales in Australian coastal waters

Kimberlee Baldry[1,2,*] , Nick Hardman-Mountford[2] , Jim Greenwood[2]

[1] University of Western Australia, Crawley, WA 6009, Australia

[2] CSIRO Oceans & Atmosphere, Floreat, WA 6913, Australia

* Current Affiliation: Red Sea Research Center, King Abdulllah University of Science and Technology, Thuwal 23955-6900, Saudi Arabia

*Correspondence to*: Kimberlee Baldry (baldry.kimberlee@gmail.com, kimberlee.baldry@kaust.edu.sa)



**Abstract.**

Owing to a lack of resources, tools, and knowledge, the natural variability and distribution of Total Alkalinity (TA) has been poorly characterised in coastal waters globally, yet variability is known to be high in coastal regions due to the complex interactions of oceanographic, biotic, and terrestrially-influenced processes. This is a particularly challenging task for the vast Australian coastline, however, it is also this vastness that demands attention in the face of ocean acidification (OA). Australian coastal waters have high biodiversity and endemism, and are home to large areas of coral reef, including the Great Barrier Reef, the largest coral reef system in the world. Ocean acidification threatens calcifying marine organisms by hindering calcification rates, threatening the structural integrity of coral reefs and other ecosystems. Tracking the progression of OA in different coastal regions requires accurate knowledge of the variability in TA. Thus, estimation methods that can capture this variability at synoptic scales are needed. Multiple linear regression is a promising approach in this regard. Here, we compare a range of both simple and multiple linear regression models to the estimation of coastal TA from a range of variables, including salinity, temperature, chlorophyll-a concentration and nitrate concentration. We find that regionally parameterised models capture local variability better than more general coastal or open ocean parameterised models. The strongest contribution to model improvement came through incorporating temperature as an input variable as well as salinity. Further improvements were achieved through the incorporation of either nitrate or chlorophyll-a, with the combination of temperature, salinity, and nitrate constituting the minimum model in most cases. These results provide an approach that can be applied to satellite Earth observation and autonomous in situ platforms to improve synoptic scale estimation of TA in coastal waters.





## 1 Introduction

Ocean acidification (OA), the reduction in oceanic pH caused by the oceans' uptake of anthropogenic CO2
emissions, is a global phenomenon predicted to impact entire marine ecosystems, from microbial primary
producers to top predators (Mostofa et al., 2016). Calcifying marine organisms generally show evidence of
stress under ocean acidification scenarios, although the effects can vary widely, are species specific, and may
depend on physiological traits (Comeau et al. 2017; Edmunds et al. 2016; Azevedo et al., 2015; Jokeil, 2016).
Calcifying organisms potentially affected include corals, calcifying algae, molluscs, foraminifera, echinoderms,
crustaceans, and bryozoans. In a high $CO_2$ world, elevated ocean $CO_2$ concentrations may stimulate
photosynthesis (Gattuso et al., 2014) but may also decrease nitrogen fixation, leading to an overall decline in
primary productivity (Hong et al. 2017), with uncertain but most-likely detrimental impacts on trophic
interactions (Nagelkerken and Connell, 2015). Furthermore, the collapse of habitat builders such as corals,
coralline algae and molluscs would have destructive impacts on entire marine ecosystems. Mesocosm and
laboratory experiments have shown the majority of calcifying corals tested experience large declines in
calcification and growth under OA scenarios (Gattuso et al., 2014).

Australia's coastline is over 36,000 km long, spanning ~33 degrees of latitude, from the tropics to the Southern
Ocean. This coastline comprises unique and diverse marine ecosystems with high levels of endemism, of which
the most famous is the Great Barrier Reef, the largest coral reef system on the planet and described as one of the
seven natural wonders of the world (Mongin et al., 2016). The World Heritage-listed Ningaloo Reef system and
remote reef systems of the Kimberley and Pilbara coasts in Western Australia are other examples of Australia's
vulnerable coral habitats. Elsewhere, sponges, bryozoans, molluscs and crustaceans contribute a significant
calcifying fauna, including some commercially significant species of abalone and scallop. Tracking and
predicting the rate of progression of OA in these systems, to inform local management actions, requires the
development of robust, and cost-effective methods of monitoring the marine carbonate system at synoptic
scales.

Fundamental to understanding the rate of OA and oceanic uptake of $CO_2$ is a requirement to quantify the
buffering capacity of seawater, which is a direct function of its total alkalinity (TA). Waters with higher TA are
less prone to rapid change in ocean pH and may provide refuge for marine biodiversity in the face of OA. TA is
conservatively related to salinity due to convective mixing and the addition or removal of freshwater to a water
mass (Cai et al., 2010; Jiang et al., 2014; Lee et al., 2006; Millero et al., 1998). This relationship has been
exploited to predict alkalinity at the global scale from historical databases of ocean salinity (Lee et al., 2006;
Millero et al., 1998).  While this works well for open ocean regions, alkalinity in coastal regions can be
significantly more variable due to a wide range of freshwater end-points in the TA-salinity relationship and the
contribution of various processes that are non-conservative with salinity (e.g. dissolved organic inputs and
biological processes such as calcification and organic matter production). Thus, these relationships may not be
robust in coastal regions (Bostock et al., 2013; Cai et al., 2010).

To address this limitation, additional proxy variables have been incorporated into predictive alkalinity models to
account for processes that affect TA non-conservatively (Jiang et al., 2014). For example, seawater temperature



has been used to help account for mixing between water masses (Jiang et al., 2014; Lee et al., 2006) and associated nutrient changes. To account for primary production, chlorophyll-a (CHL) (Hales et al., 2012), dissolved oxygen or nitrate (N) have featured in such models (Bostock et al., 2013; Brewer and Goldman, 1976; Lee et al., 2008). However, there is still uncertainty about which proxies are the most robust to use, with the choice often being influenced by availability and location. In the literature, CHL is rarely used in linear regression (LR) models for TA, but rather N is included as a third explanatory variable after salinity (S) and temperature (T). This is not only because N directly affects TA, but also that several other processes that influence TA variability can co-vary with N, making it a useful proxy (Hieronymus and Walin, 2013). Nonetheless, an important factor in considering such proxies is their amenity to broad-scale measurement. While N can be measured *in situ* using UV absorption sensors deployable on autonomous platforms (Johnson and Coletti, 2002), it cannot be measured directly from satellite instruments as it lacks an electromagnetic signature (Sarangi, 2011). There are, however, well developed remote sensing algorithms for retrieval of oceanic surface CHL concentration from satellites, suggesting a possible advantage for the use of CHL as an explanatory variable over N.

There has been little investigation of the distribution of TA in Australian coastal waters due to sparse availability of measurements. Global (Lee et al., 2006; Millero et al., 1998) and regional (Lenton et al., 2015) algorithms have been applied to the oceans surrounding Australia but little progress has been made in investigating variability in TA and its drivers in Australia's coastal waters. In this paper we analyse a seven year time series of observations from nine National Reference Stations (NRS) around Australia in order to quantify the suitability of a range of conservative and non-conservative regression models to predict TA in Australian coastal waters at local and national scales. An important consideration of this work is the ability to scale up predictions to synoptic scales, thus we focus on proxy variables that can be measured from satellite Earth observation or autonomous *in situ* technologies such as gliders.





## 2 Data and Methods

### 2.1 National Reference Station (NRS) Data

As part of Australia's Integrated Marine Observing System (IMOS), time series data for nutrient and carbon variables are routinely measured at nine National Reference Stations (NRS) located around the Australian coastline (Lynch, Morello et al. 2014, Fig. 1). Measurements are made, processed, and quality controlled using consistent methods across all stations, as described by Morello, Galibert et al. (2014), and are available from the Australian Ocean Data View (AODV) portal (https://imos.aodn.org.au). For this study, time series of temperature (T), salinity (S), total alkalinity (TA), chlorophyll-a (CHL) concentration and nitrate (N) concentration were used. These measurements were made at monthly-to-quarterly frequency as some NRS were established later than others and were sampled at different frequencies. One outlier, measured at the Maria Island NRS on the 14/02/2013 at 10m, was removed (S=23.15). The sampling points used for each NRS are presented as a time series in Supplementary Fig. 1.

Sampling and measurement protocols were undertaken according to the National Reference Stations Biogeochemical Operations Handbook (2016) and the Pre-Run Check and Field Sampling CTD Procedural Guide (2014), available for download from the IMOS website (http://imos.org.au/moorings_documentation.html). Triplicate samples of TA, S, N, and CHL were collected from Niskin bottles at 10m depth intervals. TA samples were poisoned with mercury chloride solution upon collection. All samples were then taken back to the laboratory for analysis. TA was determined by an automated open cell potentiometric titration using 0.1 M HCl as the titrant. The data for S used in this work was collected from bottle salinity data measured by a Guildline Autosal 8400B salinometer using conductivity ratios. CHL data was collected from filtered phytoplankton biomass, analysed using HPLC. The phytoplankton biomass was collected over the whole water column at the sample site thus the value used is an integrated CHL value. Finally, N was measured using a Lachat 8000 flow injection analyser with detection limit of 0.1 μM. The resulting data were quality controlled (QC) and flagged according to the National Reference Stations Biogeochemical Operations handbook (2016). Only results flagged 1 or 2 were used.

Profiling SeaBird 19+ SEACAT Conductivity, Temperature, and Depth (CTD) instruments were used for continuous measurements of T, which were then binned into 1m depth intervals. Measurements were quality controlled according to the Australian National Moorings Network (ANMN) Standardised Profiling CTD Data Processing Procedures Appendix 4, using the SBE Data Processing-Win32 software and the IMOS MATLAB toolbox, before they were uploaded to the portal (procedures and toolbox available at https://github.com/aodn/imos-toolbox). In this study, only TA measurements from the upper 60 m of the water column were used, and only CTD measurements sampled within one hour and 1 m vertically of bottle sampling were used in our analyses.

### 2.2 Linear regression (LR) analysis

General and regional models were constructed from LR analysis using the four base models (BM) shown below and the lm() function in R. General models refer to those derived from a combined dataset collected from all




nine NRS. Regional models refer to those derived from data collected from singular NRS. In total, 40 models were derived (the 4 base models applied to 1 general coastal model and 9 regional models).

$$BM1: TA = aS + d$$
$$BM2: TA = aS + bT + d$$
$$BM3: TA = aS + bT + clog[CHL] + d$$
$$BM4: TA = aS + bT + clog[N] + d$$

where T is water temperature, S is salinity, CHL is chlorophyll-a concentration, N is nitrate concentration, and a-d are constants calculated via LR.

Some of the regional NRS data sets had small numbers of observations (n) for some variables, which is not ideal for LR (Table S1-S5), particularly the Ningaloo, Darwin, and Esperance NRS. For BM4, the number of observations used in LR analysis was significantly reduced at Yongala and Kangaroo Island NRS, with only four NRS possessing a robust number of observations (n ≥ 30*[number of explanatory variables]). The results of these models are still presented although they should be considered to be less robust than those for stations with higher n. For the combined data set, Shapro-Wilk normality tests rejected the hypothesis that S, T, log[N], and log[CHL] were each individually normally distributed. It is rare for such data to resemble a normal distribution closely and it was concluded that the symmetrical distributions of S, T log[CHL], and log[N] were acceptable to proceed with LR analysis.

All residuals showed evidence of being normally distributed, appearing trendless, and homoscedastic, as should be seen for LR. Some quantile-quantile (Q-Q) plots (not presented) showed evidence of outliers, however the decision was made not to remove these apparent outliers due to the small size of some data sets.

**2.2 Open ocean model (Lee *et al.* 2006)**

In order to compare the performance of the models tested with an open ocean 'base' model, TA was reconstructed from an implementation of the model of Lee et al. (2006) using observed S and T measurements collected at the nine NRS. The open ocean model is a quadratic model and has one dynamic geographical boundary through Australian coastal waters, which varies seasonally with T. Like BM2, the number of observations able to be modelled by the open ocean model was restricted by temperature. The numbers of observations used for the open ocean model are given in Supplementary Table 2.

**2.2 Statistical analysis**

Three statistical measures and one test were utilised in order to compare models, assess their robustness and determine the minimum model, which is the model that minimises information loss from the observations.

1. **Residual standard error (RSE)** was calculated as a measure of the error in a model, when compared to observations. By multiplying by the appropriate standard z-value, 1.96 from the standard normal distribution, we obtain an approximation of the 95% confidence error (CE) associated with the model. These estimates are not reliable for models with n < 30, which will have a larger CE in accordance with the central limit theorem.



2. Bootstrapped **Kolmogorov–Smirnov (KS) tests** were employed in order to test the hypothesis that reconstructed alkalinity values are drawn from the same distribution as observations. These were tested at a 5% significance level. As both data sets in the KS tests came from the same environment (same sample of water) the test had to be bootstrapped (Kleijen, 1999). P-values are shown in Supplementary Table 5.

3. The **Akaike Information Criterion (AIC)** measures the relative quality of statistical models and is particularly useful when models with different numbers of variables are being compared. In calculating AIC there is a trade-off between the goodness-of-fit and the complexity of the model, adding an extra level of analysis compared to RSE. Using AIC values, the **relative probability of minimising information loss (RPMIL)** for each model can be determined, which allows a more intuitive and robust method for comparing models, allowing the minimum model to be determined with some level of certainty.





## 3. Results

### 3.1 Open ocean model (Lee *et al.* 2006)

Figure 2 shows the differences between modelled and *in situ* TA observations using the open ocean model at the nine NRS. All nine NRS showed RSE less than 14 µmolkg$^{-1}$. The model performed particularly well at the

Kangaroo Island NRS, predicting TA with an average difference of - 0.70 µmolkg$^{-1}$ (i.e. lower than *in- situ* observations) with a residual standard error (RSE) of 5.40 µmolkg$^{-1}$. However, the model underperformed significantly at the Darwin and Yongala NRS, while also overestimating at the remaining six NRS. At the Darwin NRS, on average the model predicted TA to be 20.28 µmolkg$^{-1}$ lower than observed values while at the Yongala NRS on average the model predicted TA of 14.65 µmolkg$^{-1}$ above observed values.

### 3.2 Kolgorov-Smirnov (KS) tests

Table 1 shows results of KS tests between respective models and observed TA with a 95% confidence level. The statistical distribution of TA was only successfully modelled for all NRS using regionally developed algorithms that include N, T and S, and not by general models for all Australian waters. Nonetheless, regional models that only use S were also able to significantly reproduce the statistical distribution of TA, with the exception of the

North Stradbroke Island, Maria Island, and Yongala NRS. At a regional level, observations from the Maria Island and North Stradbroke Island NRS were successfully modelled with BM2, BM3, and BM4, but the Yongala NRS was only successfully modelled with BM4. All NRS that were successfully modelled regionally by BM1, were also successfully modelled regionally by all other base models.

### 3.3 95% Confidence Errors (CE)

95% CE are shown in Fig. 3. The combined general model showed a marked decrease in error over BM1-BM2, and comparable errors over BM2-BM4. Regionally, most NRS exhibited similar errors over the four base models, with the exceptions being Darwin and Ningaloo. The Darwin NRS showed particularly high errors over the 4 base models. Lowest errors were given by BM3 (Darwin, Esperance, Ningaloo, North Stradbroke Island, Port Hacking Bay, Yongala) or BM4 (Kangaroo Island, Maria Island, general coastal). Overall, errors were

highest for the Darwin and Yongala regional models, and the general coastal models, with 95% CE >10 µmolkg$^{-1}$. All other models had 95% CE < 10 µmolkg$^{-1}$ for BM2-BM4.

### 3.4 AIC and RPMIL

AIC values and RPMIL are displayed in Fig. 4 and Fig. 5 respectively. AIC values are clearly higher for BM1 in all cases. Little difference in AIC is seen between BM2 and BM3. AIC values indicate that BM4 is clearly the

minimum model, with the exception of the Port Hacking Bay NRS, which shows similar AIC values at BM3 and BM4. When transforming AIC values to RPMIL, a higher value indicates a higher probability of that particular model representing the minimum model. Results are presented with the exclusion of BM4 as this is obviously the minimum model, having the lowest AIC at all stations except Port Hacking Bay NRS. At Port Hacking Bay, BM3 is the minimum model with BM4 having a RPMIL of 0.08 (ST6). Notably of BM1-BM3,

BM3 has the highest RPMIL for all models.



**4 Discussion**

This paper presents a comparison of different linear regression (LR) models for the prediction of alkalinity in Australian coastal waters. We show that including not only salinity (S) but also temperature (T), and either chlorophyll (CHL) or nitrate (N) concentration in these models can significantly improve their performance. In other regions, a simple linear TA-S dependence has often been assumed when estimating TA for use in calculations of other carbonate parameters (Bates et al., 2006; Hales et al., 2012; Lee et al., 2008; Majkut et al., 2014). In Australian waters, this approach has also been utilised at a continental scale (Lenton et al., 2015; Takahashi, T., et al. 2014; Lenton et al., 2012), but not yet at a regional scale within coastal waters. Despite the wide application of such regression approaches in estimating TA, little investigation has been undertaken on the sensitivity of TA estimates to different input variables in coastal waters. This is surprising given the wider range of processes that can influence TA in coastal waters beyond a simple water-mass mixing model, such as variable inputs of nutrients and dissolved organic material, and their influence on primary production.

LR is well recognised as a useful predictive tool for spatial extrapolation, particularly in comparison to neural networks which are proven to have less predictive power in extrapolation (Lefèvre et al., 2008). Given the goal of enabling predictions of TA in areas of sparse *in situ* measurements, we restricted the range of input variables in three base models (BM) to those available with broad coverage from satellite Earth observation, namely T, S, and CHL as a proxy for processes related to primary production (including nutrient contributions to TA). Additionally, a fourth BM that included nitrate (N) rather than CHL was included for comparison. We included nitrate, due to its direct effect on TA, as well as its co-variation with other nutrients which directly affect TA.

A log transformation was applied to CHL to account for its well-described, log-normal distribution in the ocean (Campbell et al., 1998) and to satisfy the normality assumption of LR analysis. The same transformation is applied to N as it was strongly right skewed. The technology does not currently exist to remotely sense nitrate from satellites, so BM4 is not useful when considering algorithms that can be applied to Earth observation. Nonetheless, BM4 can be utilised with data from autonomous platforms equipped with nitrate sensors, such as gliders and biogeochemical profiling floats, thus was regarded important to include in this work. Nitrate levels in some ecosystems, such as coral reefs, can often be lower than detection limits. This was a challenge with the data set used in this study and provides another limitation to BM4.

It was found from AIC values that using S alone as a predictor for TA does not give the most informative results, and that the addition of T to the model substantially increases the information of the model. This is found at all NRS locations, and between the general and regional models. This conclusion is not as strongly reflected in model errors (Fig 2) due to the substantial decreases in observations seen between the four models, and such AIC values are important to consider within model comparisons. Thus we recommend that as a minimum, T be included in regression models for the estimation of TA in Australian coastal waters.

Further, AIC values indicated that the addition of a third variable increased the information of the model, however, the decrease was less in comparison to the decreases observed when adding T as a predictor. BM4 was the minimum model for estimating TA in Australian coastal waters, at all stations except North Stradbroke





Island, where BM3 was the minimum model. Probabilistically, the addition of another predictor does add information (decreases in AIC observed over all NRS locations). However, BM4 was shown to increase model errors at 5 NRS (Darwin, Esperance, North Stradbroke Island, Rottnest Island, and Yongala) even though BM4 was the minimum model at these locations. This result is likely due to large differences between the numbers of observations available for BM4 and BM3 at these NRS, which increased RSE values in BM4.

KS tests showed that regionally-modelled TA values, constructed from BM4, were statistically similar to NRS observations, at all stations. It should be noted that although the number of data points was low for some of these stations, this should not affect the result of the general model KS tests. Although BM3 was the minimum model at the North Stradbroke Island NRS, KS tests indicated that BM4 still produced values close to observed TA at this location. Thus the addition of another predictor is advantageous in modelling TA in Australian coastal waters, with the addition of N achieving better results over CHL.

A major finding relates to the use of globally-parameterised open ocean algorithms for modelling TA. It has been shown that such algorithms often fail in coastal waters due to the strong influence that coastal processes have on the distribution of TA (Bostock et al., 2013;Cai et al., 2010). Our results confirmed that such open ocean models are not necessarily optimal for predicting TA in coastal waters and their use can result in large systematic errors in some regions (Fig. 2). Nonetheless, the use of a global model at the Kangaroo Island NRS appears to be consistent with regional parameterisations, and is further supported by KS tests. KS tests also suggest that the Lee *et al.* (2006) algorithm performs reasonably at Ningaloo. However, this result is due largely to the low number of observations obtained at the Ningaloo NRS and the large amount of scatter in observations, which reduces the sensitivity of the result. Examination of RSE values shows systematic error (bias) is introduced when the open ocean model is used.

Although KS tests show that TA is best modelled regionally, the general coastal models could still be used effectively. The general coastal models do not perform well at some locations (Table 1), but generally low RSE was observed when these models included temperature. The general coastal model, when T was included, achieved RSE < 9 $\mu$mol kg$^{-1}$. This RSE was comparable to using the Lee *et al.* (2006) model (8.6 $\mu$molkg$^{-1}$) when temperature was included, thus the general coastal models will be able to be used with a greater level of confidence for Australian coastal waters as they eliminate systematic biases.

Yongala NRS was the only NRS not successfully modelled regionally by BM3 according to KS tests. When looking at residuals, a large seasonality component is unaccounted for by BM3. The Yongala NRS is located next to the Great Barrier Reef, so it seems plausible that the extra seasonality component seen in the data could be attributed to calcification processes or benthic biological nitrogen uptake on the reef. The mouth of the Burdekin River is also located close to the Yongala NRS, so the residuals could also be capturing a changing freshwater endpoint in the model as the riverine input changed seasonally. Indeed, the seasonality in the residuals was seen to co-vary with seasonal freshening in salinity data (SF2). All of these processes affect TA signals directly, so a clear signal may be seen in the data (Brewer and Goldman, 1976; Cai et al., 2010; Jiang et al., 2014; Shadwick et al., 2011), but cannot be monitored via remote sensing (or BM3). In fact, when N was





used instead of CHL (BM4), the seasonality disappeared and was replaced by a more random distribution of residuals, although it should be noted that the data here was sparse (n=56). Additionally, when the residuals of BM3 were plotted alongside salinity, there is clear evidence of co-variability which points towards a changing riverine endpoint at that location. Thus, the N at the Yongala NRS is a proxy for riverine alkalinity inputs , which is not represented by CHL. This example both highlights a challenge faced when considering the future of remotely sensed TA, and adds additional justification that BM4 is the best model for predicting TA out of the four models tested.

The regional dependence of regression relationships for the prediction of TA in this study highlights a large limitation to broad-scale predictions of the progression of ocean acidification in vulnerable coastal regions, namely the paucity of high quality TA observations available for development of suitable algorithms. Australia has benefited from the establishment of national reference stations as part of an Integrated Marine Observing System (IMOS) that takes consistent time series observations around the coast. Nonetheless, TA observations have only been collected since 2009, so the temporal range of this data is minimal. Spatially, the data is also limited to only nine locations around a 36,000 km long coastline. For the Ningaloo, Darwin, Kangaroo Island, and Esperance stations, the number of available observations was particularly low resulting in models for these locations being statistically less robust. The Ningaloo and Esperance NRS were removed in 2015 due to budget constraints, removing the opportunity for extending these relationships in future (Note also that this leaves only one reference station monitoring the western third of Australia's coastal environment). For many parts of the world, even this level of observation is not currently achievable, increasing the challenges of monitoring the progress and impacts of ocean acidification over coming decades.

A promising opportunity lies in the application of regional relationships to satellite Earth observation data, a direction that so far has been investigated little. Recent advances in Earth observation mean that salinity, temperature, and chlorophyll-a are able to be remotely sensed using a range of passive (visible spectrum radiometry) and active (microwave and radar) sensors on orbital satellites (Land et al., 2015). This opens up avenues for exploitation of LR models developed from in situ data to enable synoptic-scale monitoring of TA variability and other carbonate system parameters. While such approaches have been successfully trialled for open oceans (Lee et al., 2006; Millero et al., 1998), less effort has been invested on its application at the coastal scale. The success of this application will depend largely on the resolution of the satellite data that the algorithm is applied to, the accuracy of the algorithm itself and the ability to quantify associated errors, increasing the need for high quality *in situ* measurements. Satellite observations are vulnerable to inaccuracies in coastal waters due to factors including cloud cover, the presence of coloured dissolved organic matter (CDOM) and suspended sediments, the presence of both marine and terrestrial aerosols, land adjacency effects, and the electromagnetic complexity of coastal signals (in both optical and radio wave spectrum) (Schalles, 2006;Land et al., 2015). Future planned sensors with higher spatial and spectral resolution may help reduce these current limitations. For illustrative purposes only, therefore, we have used inputs from satellite remote sensing to make a first estimate for average TA at the scale of the Australian continent using the general coastal model for BM3 (Figure 6).



The chemistry of the ocean is dynamic and varies between seasons and years, as well as through direct uptake of anthropogenic $CO_2$ emissions, and the influence of changing water temperature and salinity from climate forcing. Empirically-parameterised algorithms for TA may therefore require regular retuning to remain robust through time. The presence of ocean acidification will change TA through increasing carbonate dissolution over time (Cross et al., 2013), a process which cannot be estimated from any of the proxy variables explored in this paper. This might change the required algorithm inputs significantly and increase uncertainties in algorithms over time. As such, on-going *in situ* monitoring for alkalinity and other carbonate system parameters will continue to be required to support synoptic scale approaches to monitoring the progression of ocean acidification.



## 5 Conclusion

In addressing the two main applications of the results of this paper, we have defined two different minimum sets of variables for the prediction of TA in coastal waters: S, T, and log[CHL] for applications to satellite Earth observations, and S, T, and log[N] for *in situ* applications. We find that the influence of biological responses on the distribution of TA can be significant at some locations in Australian coastal waters, and must be considered when estimating TA. Finally, we have shown that when predicting TA from ocean observations, the use of T as a predictor will improve the model significantly and the addition of a third predictor offers further improvement. With this information and the models presented in this paper, more informed decisions can be made about modelling TA in Australian and other coastal waters, assisting efforts to track the progress of ocean acidification.





## Code Availability

No special code was required to model the data, and data processing can easily be followed by the methodology given. Thus no code is required to accompany this publication but can be requested.

## Data Availability

Open access at https://imos.aodn.org.au.

## Team List

Authors: Kimberlee Baldry[1,2] , Nick Hardman-Mountford[2] , Jim Greenwood[2]

Other contributors: Francois Dufois[2] , Bronte Tilbrook[3]

[1] University of Western Australia, Crawley, WA 6009, Australia

[2] CSIRO Oceans & Atmosphere, Floreat, WA 6913, Australia

[3] CSIRO Oceans & Atmosphere, Hobart, TAS 7001, Australia

## Author Contribution

K. Baldry performed all data processing and analysis. The planning of the work was performed by K. Baldry and N. Hardman-Mountford. N. Hardman-Mountford and J. Greenwood provided background knowledge,
conceptual and technical inputs and corrections. The paper was written by K. Baldry with inputs from all co-authors.

## Competing Interests

The authors declare that they have no conflict of interest.


## Acknowledgements

Thank you to the CSIRO Vacation Program for supporting this research and also SGS Australia for the receipt of the Brian Doran Scholarship for Physical Chemistry during the period of research. We thank Francois Dufois for producing Figure 6. We acknowledge the immense effort of Australia's Integrated Marine Observing System
(IMOS) in collecting the observations used for this study, especially the ocean carbon monitoring team under the leadership of Bronte Tilbrook, and the continued effort of NASA and the Barcelona Expert Center to provide EO data products for open use.





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



**Figures**

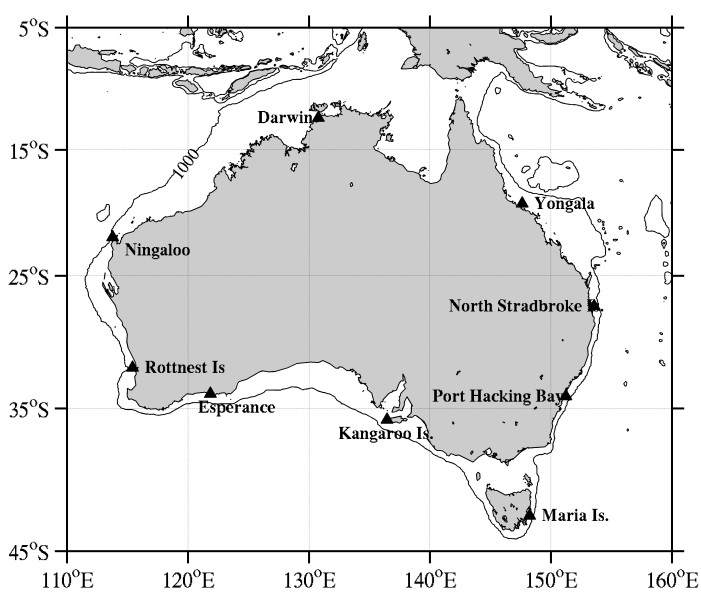

**Figure 1: Map displaying the positions of nine national reference stations (NRS)**



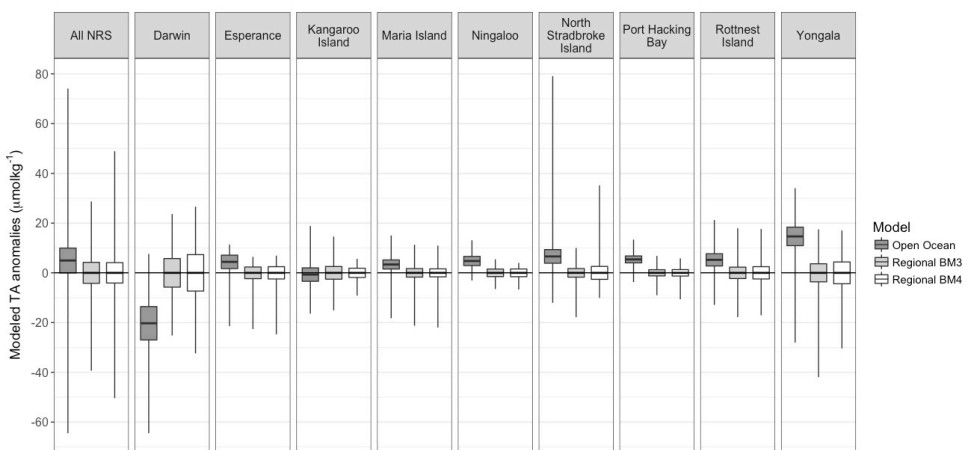

**Figure 2: Plots of the Residual TA measurements for the nine NRS, as calculated by subtracting observed TA values from those modelled by the open ocean model of Lee et al. (2006). The mean residual is shown as a black line the height of the box corresponds to one standard deviation, and the extremities of the whiskers show maxima and minima.**

25



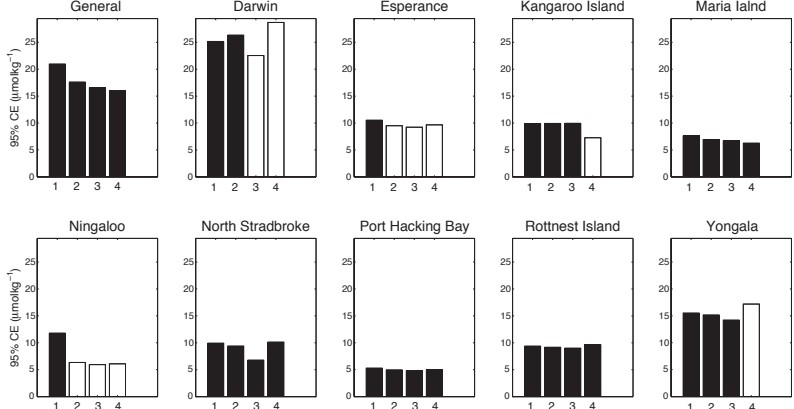

**Figure 3: 95% Confidence Errors for each of the four models tested at the coastal level and regional level for the nine NRS. Hollow bars indicate results obtained from algorithms developed from a low number of observations.**

25

30





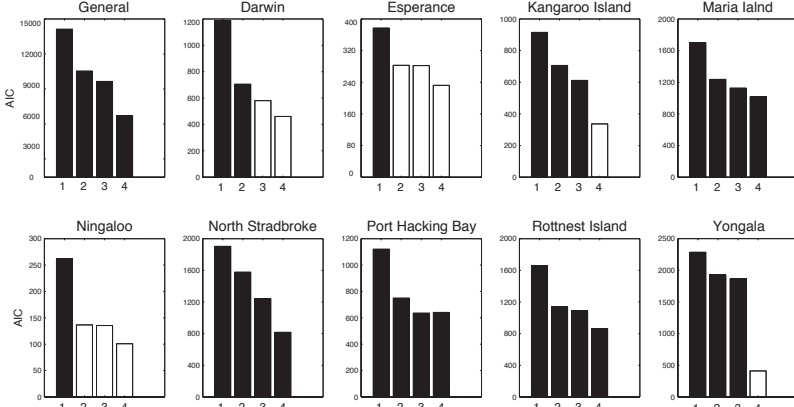

**Figure 4: AIC values for each of the four models tested at the coastal level and regional level for the nine NRS. Hollow bars indicate results obtained from algorithms developed from a low number of observations.**

25




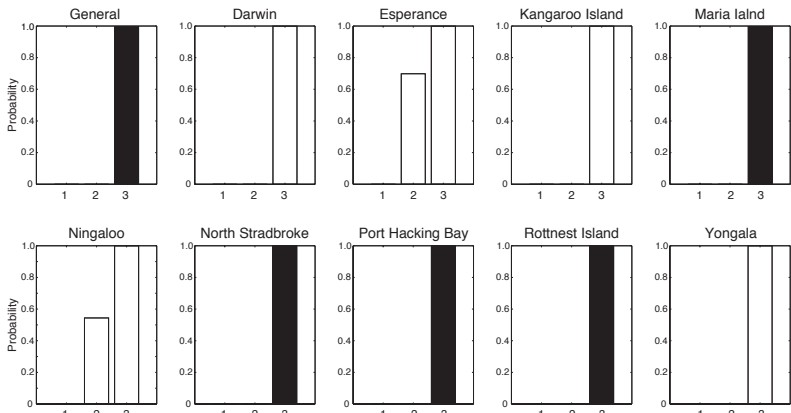

**Figure 5: Relative probabilities of minimising information loss for each of the four models tested at the
coastal level and regional level for the nine NRS. Hollow bars indicate results obtained from algorithms
developed from a low number of observations.**




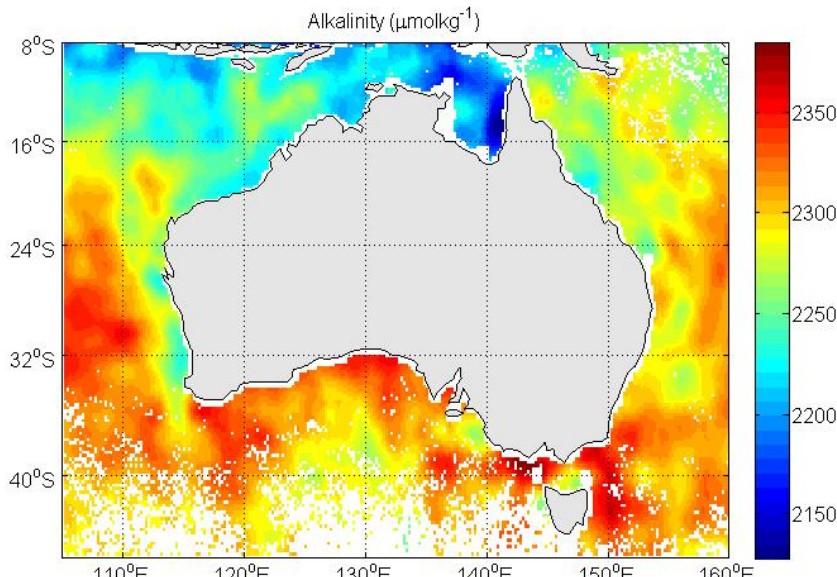

**Figure 6: Average distribution of TA around the Australian coast for May 2015, based on the extrapolation of the general regression of BM3. The figure was constructed using the general model presented in Table S3 implemented with satellite-derived MODIS 4km monthly averaged sea surface temperature, MODIS 4km monthly averaged chlorophyll-a (NASA Earth Data portal: https://earthdata.nasa.gov/) and SMOS 0.25 degree monthly averaged sea surface salinity (Barcelona Expert Center data portal: http://cp34-bec.cmima.csic.es/) data products.**





# Tables

**Table 1: Results for the K-S tests for differences in distribution.**
**✓ = statistically similar to observations**
5 **✗ = statistically different to observations**
**\* algorithms developed with low numbers of observations employed (n < 30xnumber of explanatory variables)**

| NRS | Base Model 1 | | Base Model 2 | | Base Model 3 | | Base Model 4 | | Lee et al. (2006) |
|---|---|---|---|---|---|---|---|---|---|
| | Regional | General | Regional | General | Regional | General | Regional | General | |
| **Darwin** | ✓ | ✗ | ✓ | ✗ | ✓* | ✗ | ✓* | ✗ | ✗ |
| **Esperance** | ✓ | ✗ | ✓* | ✓ | ✓* | ✓ | ✓* | ✓ | ✗ |
| **Kangaroo Island** | ✓ | ✗ | ✓ | ✗ | ✓ | ✗ | ✓* | ✗ | ✓ |
| **Maria Island** | ✗ | ✗ | ✓ | ✓ | ✓ | ✓ | ✓ | ✓ | ✗ |
| **Ningaloo** | ✓ | ✗ | ✓* | ✓ | ✓* | ✓ | ✓* | ✓ | ✓ |
| **North Stradbroke Island** | ✗ | ✗ | ✓ | ✗ | ✓ | ✗ | ✓ | ✓ | ✗ |
| **Port Hacking Bay** | ✓ | ✗ | ✓ | ✓ | ✓ | ✓ | ✓ | ✓ | ✗ |
| **Rottnest Island** | ✓ | ✓ | ✓ | ✗ | ✓ | ✗ | ✓ | ✓ | ✗ |
| **Yongala** | ✗ | ✗ | ✗ | ✗ | ✗ | ✗ | ✓* | ✗ | ✗ |

25