# Peer review of "Estimating total alkalinity for coastal ocean acidification monitoring at regional to continental scales in Australian coastal waters"

_Biogeosciences, 2017_

## Referee Comment (RC1) · C.W. Hunt (Referee) · 26 Jun 2017

Review of Baldry et al., "Estimating total alkalinity for coastal ocean acidification monitoring at regional to continental scales in Australian coastal waters"

Summary This manuscript uses in-situ total alkalinity and physical/biogeochemical measurements (temperature, salinity, chlorophyll fluorescence, and nitrogen) to both test a global equation for alkalinity retrievals (Lee et al. 2006) and to develop localized equations for nine coastal sites. I think this topic is potentially of significant interest to others, and I think the authors have a very nice data set to exploit. However, I think the analysis is somewhat superficial, and I finished the paper wanted a lot more analysis

and discussion that what is provided. I will address these points below, but I encourage the authors to further expand their work. The paper is generally well-written, but I list some specific comments which may improve readability.

Major Comments My major reservation about this work is the depth of analysis. This mostly relates to the results from the nine study sites. As noted in the manuscript, these sites are scattered along a very long coastline, distributed across a wide range of latitudes, and presumably represent contrasting conditions from the interactions of offshore ocean water and unique terrestrial and estuarine inputs and transformations. However, except for the discussion of the Yongala site (where the regression results were weakest), little to no information is presented to describe how these sites differ. The one citation referencing the sites (Page 5 Line 5, "Lynch, Morello et al. 2014) appears to be missing from the References. The reason I am left wanting more information about the physical settings is that a number of the regression coefficients are quite similar. By eye, it seems that under Base Model 2 the sites Kangaroo Island, Maria Island, and North Stradbroke Island have nearly the same regression line: is this true? If so, is this coincidence, or are there commonalities between these sites that might explain their similar results? Interestingly, the Ningaloo and Port Hacking Bay Base Model 2 results seem similar to each other, and they are on opposite sides of the continent! Considering Base Model 4, one might group Kangaroo Island, Ningaloo, and Rottnest Island, which is a much different cohort. Is there a statistical way to cluster the sites together to look for spatial trends? The authors describe the site-by-site results as 'regional', but how far might that region extend around each site? How can these results be applied to locations between the study sites? One recent paper by Carter et al. discusses a method for interpolating alkalinity data between station which may be helpful. Again, some understanding of what makes the study sites alike or different would help me understand how applicable these equations may be in other places. I also think the authors should read the paper by Alin et al. (2012), which may provide more insight for this work. Those authors also used multiple linear regression techniques to model alkalinity (and other carbonate system parameters) from

physical/biogeochemical data at coastal sites.

Specific Comments P2L9-10 the word "threatens/threatened" is repeated in one sentence. P2L12 what are the synoptic scales of interest? P3L2- Define CO2 (and format the subscript) P3L21-22- the phrase "contribute a significant calcifying fauna" seems pretty awkward P3L26- This line is also pretty awkward P3L29-30- "TA is conservatively related to salinity"- this is an overstatement P3L36- don't forget organic matter respiration too P4L24- again, what are the synoptic scales of interest? P5L23-24- how useful is an integrated phytoplankton biomass over the entire water column, if discrete alkalinity/salinity/temperature pairs are used? Will this affect the statistics, if the same CHL value is used for multiple alkalinity samples? This seems risky. P5 and P6: the Sections "Linear Regression (LR) Analysis","Open ocean model", and "Statistical analysis" are all numbered 2.2- shouldn't they be 2.2, 2.3, and 2.4? P6L3- these equations are listed with little in the way of introduction. Can the authors set them up more in the text before listing them? P7L10- the term "minimum model" is a little confusing to me. It makes me think this is the minimum set of input parameters needed to accurately estimate alkalinity. Perhaps this is a statistical term I am not familiar enough with, but it seems the minimum model is just the one with the lowest AIC numbers- correct? But it may be perfectly reasonable to still use the other regression models, depending on the input data available and the user's goals. P8- This section is so brief, it feels somewhat like an afterthought. Have the authors considered combining this with the next section into a dual Results and Discussion section? This might result in better flow from topic to topic. P9L10- again, the paper by Alin et al. (2012) and other related works undermine this argument that little coastal regression work has been done. P9L14-29- much of this material seems like it should be in the Introduction or perhaps Data and Methods sections- it seems out of place here. P9L39- are the "decreases" described here decreasing AIC values? Unclear from the text. P10L37-40. This argument seems a little shaky. Seasonality in river discharge would also result in a salinity seasonality as well, not just in alkalinity. Are the authors implying that the alkalinity concentrations in river discharge vary seasonally? If so, what is the mechanism for this? Also, the authors

state that this seasonality cannot be measured by remote sensing, but isn't one of the prospects held out by this paper the potential of new remotely-sensed salinity products? Why would remotely sensed salinity not show this seasonality? P11L18- change to "in the future" P11L23-38- Figure 6 might undermine several of this paper's points. For one, it only uses the general coastal model, since the authors did not attempt to interpolate their regional results over the entire domain. Also, Figure 6 shows that the actual locations of the stations where data were collected are masked out by a land mask. How sure are the authors that applying this coastal model is appropriate, given that the model data were collected in an area where remotely sensed measurements are not even available. I acknowledge that it is good to show the potential application of these relationships, but this raises the question of site selection in this study. P13L2- Again, I am confused about the use of the term "minimum" Previously the authors state that at least temperature should be included with salinity in these equations. However, this seems to contradict that recommendation by saying that the inclusion of salinity, temperature, and CHL or N are the minimum. Also, can the authors quantify, perhaps in terms of umol alkalinity error, how much better it is to include CHL or N? Figure 1- what is the inclusion of the 1000m isobath intended to show? Why 1000m? Figure 2- the text in this figure is very small, and might not show up well in the final version Figure 3 and Figure 5- please insert a space into the $\mu$mol kg-1 labels Figure 5- the caption says results from four models are shown, when only three are shown Table 1- I'd appreciate another table, perhaps describing the salinity, temperature, alkalinity, CHL and N data for each site. Perhaps just some basic statistics such as range, mean, standard deviation etc. Figure S1- Might this be better shown in a table? Or at least could this information be briefly described in the text? This figure seems a bit superfluous to include. Table S1-S4- while these tables contain a lot of information, they are also really the heart of the paper's analysis. Seems a little strange to bury them in the Supplementary Information. Also, do the terms Intercept, S, T etc. in these tables refer to the terms d, a, b respectively in the model equations listed on Page 6 of the main text? If so, please use consistent terms between the two.

Alin, S. R., R. A. Feely, A. G. Dickson, J. M. Hernández-Ayón, L. W. Juranek, M. D. Ohman, and R. Goericke. 2012. Robust empirical relationships for estimating the carbonate system in the southern California Current System and application to CalCOFI hydrographic cruise data (2005–2011). J. Geophys. Res. 117: C05033. doi:10.1029/2011JC007511 Carter, B. R., N. L. Williams, A. R. Gray, and R. A. Feely. 2016. Locally interpolated alkalinity regression for global alkalinity estimation. Limnol. Oceanogr. Methods 14: 268–277. doi:10.1002/lom3.10087

Please also note the supplement to this comment:
https://www.biogeosciences-discuss.net/bg-2017-221/bg-2017-221-RC1-supplement.pdf

---

## Referee Comment (RC2) · Anonymous Referee #2 · 7 Sep 2017

Review of the paper "Estimating total alkalinity for coastal ocean acidification monitoring at regional to continental scales in Australian coastal waters" by Kimberlee Baldry, Nick Hardman-Mountford, and Jim Greenwood, submitted to Biogeosciences for possible publication.

The manuscript "Estimating total alkalinity for coastal ocean acidification monitoring at regional to continental scales in Australian coastal waters" by K. Baldry and co-workers reports and discusses an approach to estimate total alkalinity (TA) in coastal waters with the intention to characterize vulnerability or resilience of such waters with respect to ocean acidification. The authors employ field data of TA and of further ocean water

properties to derive a suite of models to parameterize TA. In summary I have serious concerns and reservations with the paper, such that I unfortunately cannot recommend publication of the paper at its present state. I do see some potential to improve the paper, however this would require a major overhaul of the paper. I hope the authors could make some use of my comments in order to do so.

1: In fact the paper stops, where it should start. As far as I understood the paper, the paper only takes TA and related properties to derive a suite of algorithms/models to re(!)compute TA. The following discussion then compares the TA computation with the observed TA, but nothing goes beyond the use to the variables, which have been used to train the regressions. Thus, there is no estimation of TA, so far it appears to be a recompilation only. I was searching for some time for the application of these regressions, which goes beyond training, and eventually discovered a 2-line statement about figure 6 – which is for illustrative purposes only? Frankly, what is the usefulness of a colorful figure for illustrative purposes?

In my view this is the point where the paper should start, including detailed validation with respect to data, including data, which have not been used to train the regressions. In essence, anything prior to figure 6 is an extended methods section.

2: The application of the newly obtained regressions to independent data is particularly relevant to such an approach, as the causal relationship between TA and the regression properties is not clear or even not given. The extrapolation of pure empirical relationships, i.e., regression coefficients, bears the massive risk, that these only hold true within their framework of training, or trained data. It might well be the case that the extrapolation does work very well, it could also be the opposite. Figure 6 should have been the first step to open this discussion.

Along these lines the justification or even explanation of regression parameters falls short, specifically with respect to the non-conservative parameters:

A: Amongst the most powerful characteristics of TA is its temperature INDEPEN-

DENCE. Open ocean TA vs. temperature relationships are not much more than masked TA-depth relationships, if at all. If you refer to temperature as (partial) TA proxy, please explain and justify, why it is used. What about seasonality, and which processes does such a relationship mimic?

B: In a similar manner the used of Chla as partial TA proxy should be discussed. What thought does support this? Why are water column inventories used rather than actual concentrations? The satellites do not sense the water column inventory of Chla, they "see" the upper most layer? Also an important point to be considered here are the problems of remotely sensed Chla values in coastal waters (case 1 vs case 2 waters). The authors mention initially that Australia' coastline spans 33degrees in latitude, which likely causes vastly different organic matter composition of such coastal waters.

C: The use of nitrate should be justified here as well. What does it stand for, maybe as runoiff proxy, or proxy for biological metabolism?

---

## Author Comment (AC1) · 9 Oct 2017

[revised manuscript text omitted]

Understanding and quantifying distributions of total alkalinity (TA), the proton deficit of seawater relative to neutrality, is an indication of how much carbon dioxide seawater can hold. Waters with higher TA are less prone to rapid change in ocean pH, as they have a higher proton deficit to "consume" protons generated from $CO_2$ uptake, potentially offering refuge for marine biodiversity in the face of OA. Thus, TA is fundamental to understanding the rate of OA and oceanic uptake of $CO_2$. Salinity is a conservative tracer within a water mass, meaning that it only experiences changes due to mixing of different water masses or through the addition or removal of freshwater. This leads to a linear relationship between salinity and TA in a region where convective mixing occurs between two waters with differing TA signals (Cai et al., 2010; Jiang et al., 2014; Lee et al., 2006; Millero et al., 1998). This relationship has been exploited to predict alkalinity at the global scale from historical databases of ocean salinity (Lee et al., 2006; Millero et al., 1998). While this works well for open ocean regions, alkalinity in coastal regions can be more variable due to dynamic freshwater end-points in the TA-salinity relationship, the mixing of multiple oceanic end members, and the contribution of various processes that are non-conservative with salinity (e.g. dissolved organic inputs, organic matter respiration, and biological processes such as calcification and organic matter production) (Fig. 1). Thus, with the additional consideration

of non-conservative variability and convective mixing processes, it is clear that these relationships may not be robust in coastal regions (Bostock et al., 2013; Cai et al., 2010).

[revised manuscript text omitted]

**2.2 Linear regression (LR) analysis and model development**

LR is well recognised as a useful predictive tool for spatial extrapolation, particularly in comparison to neural networks which are proven to have less predictive power in extrapolation (Lefèvre et al., 2008). Given the goal of enabling predictions of TA in areas of sparse *in situ* measurements, we restricted the range of input variables to those available with broad coverage from satellite Earth observation, namely T, S, and CHL. Additionally, a fourth BM that included nitrate (N) rather than CHL was included for comparison, which can be measured *in situ* using autonomous sensors. This variable choice accommodates the conservative three end-member mixing model presented in Fig. 1, in addition to testing for variability due to primary production and other non-conservative coastal processes.

General and regional models for the prediction of TA were constructed from LR analysis using the four base models (BM) shown below and the lm() function in R. General models refer to those derived from a combined dataset collected from all nine NRS. Regional models refer to those derived from data collected from singular NRS. In total, 40 models were derived (the 4 base models applied to 1 general coastal model and 9 regional models).

$$BM1: TA = aS + d$$
$$BM2: TA = aS + bT + d$$
$$BM3: TA = aS + bT + c\log[CHL] + d$$
$$BM4: TA = aS + bT + c\log[N] + d$$

where T is water temperature, S is salinity, CHL is chlorophyll-a concentration, N is nitrate concentration, and a-d are constants calculated via LR. A log transformation was applied to CHL to account for its well-described, log-normal distribution in the ocean (Campbell et al., 1998) and to satisfy the normality assumption of LR analysis. The same transformation is applied to N as it was strongly right skewed.

Some of the regional NRS data sets had small numbers of observations (n) for some variables, which is not ideal for LR (Table S2-S5), particularly the Ningaloo, Darwin, and Esperance NRS. For BM4, the number of observations used in LR analysis was significantly reduced at Yongala and Kangaroo Island NRS, with only four NRS possessing a robust number of observations (n ≥ 30*[number of explanatory variables]). The results of these models are still presented although they should be considered to be less robust than those for stations with higher n. For the combined data set, Shapro-Wilk normality tests rejected the hypothesis that S, T, log[N], and log[CHL] were each individually normally distributed. It is rare for such data to resemble a normal distribution closely and it was concluded that the symmetrical distributions of S, T log[CHL], and log[N] were acceptable to proceed with LR analysis.

All residuals showed evidence of being normally distributed, appearing trendless, and homoscedastic, as should be seen for LR. Some quantile-quantile (Q-Q) plots (not presented) showed evidence of outliers, however the decision was made not to remove these apparent outliers due to the small size of some data sets.

**2.3 Open ocean model (Lee *et al.* 2006)**

In order to compare the performance of the models tested with an open ocean 'base' model, TA was reconstructed from an implementation of the model of Lee et al. (2006) using observed S and T measurements collected at the nine NRS. The open ocean model is a quadratic model and has one dynamic geographical boundary through Australian coastal waters, which varies seasonally with T. Like BM2, the number of observations able to be modelled by the open ocean model was restricted by temperature. The numbers of observations used for the open ocean model are the same for those used in the regional modelling of BM2 (Table S3).

**2.4 Statistical analysis**

Four statistical measures and one test were utilised in order to compare models, assess their robustness and determine the minimum model, which is the model that minimises information loss from the observations.

1. **Residual standard error (RSE)** was calculated as a measure of the error in a model, when compared to observations. By multiplying by the appropriate standard z-value, 1.96 from the standard normal distribution, we obtain an approximation of the 95% confidence error (CE) associated with the model. These estimates are not reliable for models with n < 30, which will have a larger CE in accordance with the central limit theorem.

2. **Mean absolute error (MAE)** was calculated as the mean of absolute residuals from each respective model. This is an important indication of how well a model captures the anomalous variations in the data which are more likely to be influenced by non-conservative processes, whilst keeping the measure comparable over data sets with different n. Outliers can obscure these values, so a visual assessment of residuals was conducted to assess if extreme residuals were characteristic anomalies (ie. occurred in groups or due to scatter), or single outliers. Only one outlier was found at the Esperance NRS, which was excluded in the calculation of MAE for models developed from Esperance NRS data.

3. Bootstrapped **Kolmogorov–Smirnov (KS) tests** were employed in order to test the hypothesis that reconstructed alkalinity values are drawn from the same distribution as observations. These were tested at a 5% significance level. As both data sets in the KS tests came from the same environment (same sample of water) the test had to be bootstrapped (Kleijen, 1999). P-values are shown in Supplementary Table 5.

4. The **Akaike Information Criterion (AIC)** measures the relative quality of statistical models and is particularly useful when models with different numbers of variables are being compared. In calculating AIC there is a trade-off between the goodness-of-fit and the complexity of the model, adding an extra level of analysis compared to RSE. The minimum model, the model that minimises information loss, can then be determined as the model with the lowest AIC value. Using AIC values, the **relative probability of minimising information loss (RPMIL)** for each model can also be determined which normalises differences in AIC according to the number of observations. This allows a more intuitive and robust method for comparing models, by determining probabilities that another model is actually the minimum model given infinite data points.

**2.5 Analysis of regional dependence for estimating TA**

To assess the regional dependence of the distribution of TA in the Australian coastal zone two analyses were performed.

1. **2-D Multi-dimensional scaling (MDS)** was employed to investigate the regional differences in ocean variables. A 2-D MDS plot was produced using the cmdscale() function in R, on a the subset of the entire NRS data set that contained no missing values for TA, S, T and CHL. N was not included in this analysis as it was shown to have a smaller variability (Table 1).

2. A network was constructed from the results of K-S tests performed (as described in Section 2.4) between models from each sites. For each base model, a matrix was constructed as below, with the number 1 indicating that observations at one NRS were significantly similar to reconstructed TA based on a regression trained from the data of a different NRS.

$$
M_{BMk} = \begin{bmatrix} x_{11} & \cdots & & \cdots & x_{19} \\ & \ddots & & & \\ \vdots & & x_{ij} & & \vdots \\ & & & \ddots & \\ x_{91} & \cdots & x_{9i} & \cdots & x_{99} \end{bmatrix}
$$

where $x_{ij} = 1$ when observed values at NRS i are statistically similar to reconstructed values from base model k, trained using data from NRS j. Finally, uni-directional links (ie. where $x_{ij} \neq x_{ji}$) were ignored as to only obtain results in which both NRS are modelled to the same standard by one base model, as a cross-validation technique for cluster identification.

**3. Results**

**3.1 Open ocean model (Lee *et al.* 2006)**

Figure 3 shows the differences between modelled and *in situ* TA observations using the open ocean model at the nine NRS. All nine NRS showed RSE less than 14 µmol kg$^{-1}$. The model performed particularly well at the Kangaroo Island NRS, predicting TA with an average difference of - 0.70 µmol kg$^{-1}$ (i.e. lower than *in situ* observations) with a residual standard error (RSE) of 5.40 µmol kg$^{-1}$. However, the model underperformed significantly at the Darwin and Yongala NRS, while also overestimating at the remaining six NRS. At the Darwin NRS, on average the model predicted TA to be 20.28 µmol kg$^{-1}$ lower than observed values while at the Yongala NRS on average the model predicted TA of 14.65 µmol kg$^{-1}$ above observed values.

**3.2 Kolgorov-Smirnov (KS) tests**

Table 2 shows results of KS tests between respective models and observed TA with a 95% confidence level. Results indicate that the open ocean model produces statistically similar results to observed values at the Kangaroo Island and Ningaloo Island NRS. The statistical distribution of TA was only successfully modelled for all NRS using regionally developed algorithms that include N or CHL, T and S, and not by general models for all Australian waters. Nonetheless, regional models that only use S were also able to significantly reproduce the statistical distribution of TA, with the exception of the North Stradbroke Island, Maria Island, and Yongala NRS. At a regional level, observations from the Maria Island and North Stradbroke Island NRS were successfully modelled with BM2, BM3, and BM4, but the Yongala NRS was only successfully modelled with BM3 and BM4. All NRS that were successfully modelled regionally by BM1, were also successfully modelled regionally by all other base models.

**3.3 95% Confidence Errors (CE)**

95% CE are shown in Fig. 4. The combined general model showed a marked decrease in error over BM1-BM2, and comparable errors over BM2-BM4. Regionally, most NRS exhibited similar errors over the four base models, with the exceptions being Darwin and Ningaloo. The Darwin NRS showed particularly high errors over the 4 base models. Lowest errors were given by BM3 (Darwin, Esperance, Ningaloo, North Stradbroke Island, Port Hacking Bay, Yongala) or BM4 (Kangaroo Island, Maria Island, general coastal). Overall, errors were highest for the Darwin and Yongala regional models, and the general coastal models, with 95% CE >10 µmol kg$^{-1}$. All other models had 95% CE < 10 µmol kg$^{-1}$ for BM2-BM4.

**3.4 AIC**

AIC values are displayed in Fig. 5, and associated relative probabilities of minimising information loss (RPMIL) are presented in Table S6. AIC values are clearly higher for BM1 in all cases. AIC values indicate that BM3 is clearly the minimum model at 5 NRS, with the exception of the Esperance, Kangaroo Island, Ningaloo and Yongala NRS for which BM4 is the minimum model. Little difference in AIC is seen between BM2 and the minimum model at the Ningaloo, Esperance and Rottnest Island NRS, although when translated to RPMIL it is clear that probabilistically, the minimum model is almost certainly (>90%) that indicated by AIC values. AIC values indicate that the minimum model for the Yongala NRS and the Maria Island NRS was not the best in

terms of RSE and MAE. Upon further analysis, at the Maria Island NRS BM3 removed two anomalous TA residuals as there was not CHL available to them. Thus BM3, the minimum model, is not the best model at this site and BM4 is a better choice.

5    **3.5 Regional dependence and clustering**

Mean and standard deviations for the studied ocean variables are presented in Table 1. A network map representing the results for cluster identification, using p-values from K-S tests as an indication of connectivity via different BM is presented in Fig. 6. A dominant cluster (C1) between three NRS was identified, containing the Rottnest Island, Esperance and Port Hacking Bay NRS, which are linked by BM1, BM2 and BM3. A second
10   cluster (C2) linked by BM2 and BM3 also was identified, which includes all members from C1, and the North Stradbroke Island NRS. This aligns well with other K-S results, which show that at the North Stradbroke Island NRS TA cannot be modelled by S alone. A final cluster (C3) was identified, connected by BM4, which includes the Esperance, Maria Island and Ningaloo NRS. The 2-D MDS plot (Fig. 7) shows a clear regional gradient, when considering selected ocean variables, and it is evident that the members of C1 and C2 are geometrically
15   close and have similar seawater characteristics. However, the members of C3 are relatively geometrically distant as indicated by median the geometric positions of each NRS as displayed on the 2-D MDS plot. Additionally, in C1 and C2 clusters members have differing minimum models, meaning that error could be introduced through the use of BM3 at NRS, which have a regional recommended model of BM4 (Esperance NRS). Regression parameters for each cluster are presented alongside the results from other models in the supplementary material.

**3.6 A special case: The Yongala NRS**

The Yongala NRS displayed a dominant inter-annual trend within the residuals of all regional BM. Upon further investigation it was revealed that this trend was a reflection of an inter-annual trend in TA, which co-varied closely with S (Fig. 8). It was also found that anomalous values were obscuring statistical tests, displaying high
25   statistical leverage on models, which alters fit. This means that on average, BM3 and BM4 actually increased the error of a large portion of residuals further indicated by increases in MAE (Table S2-5).

30

35

**4 Discussion**

This paper presents a comparison of different linear regression (LR) models for the prediction of alkalinity in Australian coastal waters. In other regions, a simple linear TA-S dependence has often been assumed when estimating TA for use in calculations of other carbonate parameters (Bates et al., 2006; Hales et al., 2012; Lee et

5    al., 2008; Majkut et al., 2014). In Australian waters, LR has also been utilised at a continental scale for the prediction of TA in carbon studies (Lenton et al., 2015; Takahashi, T., et al. 2014; Lenton et al., 2012), but not yet at a regional scale within coastal waters. Despite the wide application of such regression approaches in estimating TA, little investigation has been undertaken on the sensitivity of TA estimates to different input variables in this region. This is surprising given the wider range of processes that can influence TA in coastal

10    waters beyond a simple water-mass mixing model, such as variable inputs of nutrients and dissolved organic material, and their influence on primary production. Here we provide recommendations for the modelling of TA for carbon studies at 9 NRS (Table 3), and have shown that the inclusion of not only salinity (S) but also temperature (T), and either chlorophyll (CHL) or nitrate (N) concentration in these models can significantly improve their performance. In addition to exploring regional relationships, we explore the possibility of

15    estimating TA robustly at synoptic scales, which include the locality of multiple NRS. We find that the Esperance, Rottnest Island, North Stradbroke Island and Port Hacking Bay NRS can be combined for robust monitoring at a synoptic scale. However, regular cross-validation should be performed to identify anomalous events, or deviations from trends, as these NRS are on the opposite side of the continent and are influenced by different long-term climate patterns. This knowledge can be employed to construct cost-effective strategies for

20    the monitoring of TA and considered for application to the monitoring other carbon variables in the Australian coastal zone.

A major finding relates to the use of globally-parameterised open ocean algorithms for modelling TA. It has been shown that such algorithms often fail in coastal waters due to the strong influence that coastal processes have on the distribution of TA (Bostock et al., 2013;Cai et al., 2010). Our results confirmed that such open

25    ocean models are not necessarily optimal for predicting TA in coastal waters and their use can result in large systematic errors in some regions (Fig. 3). Nonetheless, the use of the open ocean model at the Kangaroo Island NRS appears to be consistent with regional parameterisations, and is further supported by KS tests. KS tests also suggest that the open ocean algorithm performs reasonably at Ningaloo, but still, a systematic error can be seen

30    in Fig. 3. This result is due largely to the low number of observations obtained at the Ningaloo NRS and the large amount of scatter in observations, which reduces the sensitivity of the result.

It was found from AIC values that using S alone as a predictor for TA does not give the most informative results, and that the addition of T to the model substantially increases the information of the model at regional

35    scales as well as at synoptic scales (Fig. 5). This is found at all NRS locations, and between the general and regional models. This conclusion is not as strongly reflected in model errors (Fig. 4) due to the substantial decreases in observation numbers (n) seen between the four models but is reflected in MAE values, and such AIC values are important to consider within model comparisons. Thus we recommend that as a minimum, T be included in regression models for the estimation of TA in Australian coastal waters. Further, AIC values

40    indicated that the addition of a third variable increased the information of the model. As such, BM3 was the

minimum model for estimating TA in Australian coastal at 5 of the NRS; elsewhere BM4 was the minimum model. Based on RPMIL values there are no situations in which two models were comparably as likely to be the minimum model (Table S7), however the minimum model for the Yongala and the Maria Island NRS are not the best models when RSE and MAE are further considered.

The regional dependence of models for modelling TA in Australian coastal waters is evident throughout the results. The 2-D MDS plot shows a tendency for observations from the same NRS to cluster together (Fig. 7). This indicates that the distribution of selected ocean variables of a NRS is dependent on its location, and cannot be explained by variability in any of the ocean variables studied, and require regionally derived algorithms. KS

10    tests showed that regionally-modelled TA values, constructed from BM4 and BM3, were statistically similar to NRS observations, at all locations, however modelled observations at a continental scale were not statistically similar to observations at all individual NRS locations for any of the BM. This shows that modelling TA at a continental scale is not the most robust method, and further illustrates how the addition of the third variable increases the confidence in successfully modelling TA regionally, if the dependence of TA on predictor

15    variables is unknown.

When taking a closer look at the 2-D MDS plot, there is evidence of connectivity between individual NRS, which could lead to clustering. To assess the significance of this connectivity, a network was constructed from the 9 NRS as outlined in Section 2.6 (Fig. 6). Three clusters were identified, which were all successfully

20    modelled from regressions developed from data collected at other NRS, which were members of the same cluster. Combining the results of the MDS plot and this cluster analysis, there is some doubt as to whether C3 is a viable model, as points on the MDS are geometrically distant meaning that sites do not display the same variability. Thus we recommend that C3 not be employed for the synoptic scale estimation of TA.  By taking into account all statistical results we have proposed a simplistic guide to help users of the NRS data set, or of

25    data collected in close proximity to NRS sites to understand which method of estimating TA is most robust (Table 3). However, knowledge on robustness alone is not enough to make informed decisions on the employment of environmental surrogates (Lindemayer et al. 2016), and users may wish to consider robustness trade offs for benefits such as cost reduction.

30    The results of Table 3 can be attributed to the differences in the oceanographic and benthic processes of the seven characteristic regions. The Yongala NRS and the Darwin NRS experience different distributions of the ocean variables studied compared to other NRS, according to the 2-D MDS plot. Darwin has a distinct geographical setting, located in the Beagle Gulf, between an island and the mainland. This location experiences highly variable freshwater inputs from land-masses, as well as seasonal perturbations from surrounding coral

35    reefs. The Darwin site, exhibits a lower salinity, compared to other NRS, which supports the hypothesis that it is influenced by freshwater. This variability is best captured using BM3, indicating that non-conservative TA anomalies have an annual seasonal dependence, like those seen in phytoplankton variability. The Yongala NRS also experiences large variability, and residuals show that there is a large, inter-annual seasonal component that cannot be explained fully by any on the models (Fig. 8).  The cause of this variability seems to be driven by

40    salinity, leading to the conclusion that increased freshwater input is the cause of this variability. When TA is

further plotted with salinity in a time series, this is obviously the case, and we see that this variability is reduced by a large factor using a regional BM2. This region is prone to flooding, and river flow data presented in Lough et al. (2015) demonstrates that during the study period (2009-2016), the region was transitioning from a period of high flooding frequency to low flooding frequency, consistent with the trends shown in Fig. 8. Further, the

5      maximum anomalies seen in the residuals corresponds to the worst flooding event, at the start of 2011 where three-quarters of the councils in the Australian state of Queensland were declared disaster zones. Here we only present the impacts of the flooding on employing pre-determined models in Yongala, however, understanding the impacts of this phenomenon on the carbon cycle would be an interesting consideration for future work.

10     The Ningaloo NRS also experiences large variability, as it is located over a coral reef. The remoteness of this NRS has resulted in relatively low levels of sampling effort (n), particularly CHL and N observations (Table 1). BM2 appears to capture most variability, and the further inclusion of nitrate to the model does appear to reduce error related to anomalies as inferred by RSE and MSE results, but more study of this area is required before an understanding of the mechanisms behind this can be reached. The Maria Island and Kangaroo Island NRS show

15     evidence of similar characterisation, due to their latitude and influence from the Southern Ocean, but are still separated to a degree on the 2-D MDS plot. This is not surprising as Maria Island has a higher influence of oceanic waters, as it is located closer to the shelf edge, and is influenced by variability in subsurface currents where as Kangaroo Island is driven by seasonal currents (Lynch et al. 2014). There is most likely a high influence of upwelling at the Maria Island NRS, producing high levels of primary production, as indicated by

20     higher nitrate and chlorophyll average distributions (Table 1). Although its water mass characteristics are very similar, the North Stradbroke Island NRS is not a member of the dominant cluster C1, and shows different mechanisms of variability. There is a large coastal bay in the vicinity of this NRS, which could be driving the dependence for the minimum model of TA on CHL, as waters from this region would exhibit higher CHL, however this is speculative and a more detailed study of biogeochemistry is required to confirm this hypothesis.

25     Finally, the estimation of TA at the synoptic scale using C1 and C2 is possible as the three locations have a shared influence of boundary currents that flow from North to South, promoting upwelling coastal in these three regions (Lynch et al. 2014; Jones et al. 2015). This means that the source waters are comparable, as latitude is the primary driver of biogeochemical distribution in the open ocean (Lee et al. 2006), and that coastal processes have a comparable effect at the three locations. In this context, it makes sense that Maria Island NRS is excluded

30     from C1 and C2, as here multiple currents converge and flow into the Tasman Sea, making the region unique (Jones et al. 2015).

Similarity deduced from average distributions of biogeochemical and physical ocean variables (Table 1) and latitude are not enough to predict the similarities observed between TA relationships at different NRS in coastal

35     Australian waters. For example, Kangaroo Island is on the same latitudinal gradient, and in close proximity to the Esperance NRS, yet has no connectivity to it through any of the base models. The results of this work show why TA relationships, in addition to average distributions, must be studied to characterise regions with similar TA, as although their average distribution may exhibit similar characteristics, their anomalous TA observations that have largely been influenced by coastal processes may exhibit different driving characteristics.

40     Consequently, interpolation between NRS cannot be recommended until further data is collected to increase

spatial resolution. Shipboard measurements offer an interesting resolution to this problem . Additionally low cost, autonomous carbon flux chambers may soon provide supplementary or alternative sampling to surface DIC measurements (Bastviken et al. 2015), allowing the entire carbonate system to be determined at significantly reduced cost.

The results of this study highlight a large limitation to broad-scale predictions of the progression of ocean acidification in vulnerable coastal regions, namely the paucity of high quality TA observations available for development of suitable algorithms. Australia has benefited from the establishment of national reference stations as part of an Integrated Marine Observing System (IMOS) that takes consistent time series observations around

10   the coast. Nonetheless, TA observations have only been collected since 2009, so the temporal range of this data is minimal. Spatially, the data is also limited to only nine locations around a 36,000 km long coastline. For the Ningaloo, Darwin, Kangaroo Island, and Esperance stations, the number of available observations was particularly low, resulting in models for these locations being statistically less robust. The Ningaloo and Esperance NRS were removed in 2015 due to budget constraints, removing the opportunity for extending these

15   relationships in the future (note also that this leaves only one reference station monitoring the western third of Australia's coastal environment). For many parts of the world, even this level of observation is not currently achievable, increasing the challenges of monitoring the progress and impacts of ocean acidification over coming decades.

20   A promising opportunity lies in the application of regional relationships to satellite Earth observation data, a direction that so far has been little investigated. Recent advances in Earth observation mean that salinity, temperature, and chlorophyll-a are able to be remotely sensed using a range of passive (visible spectrum radiometry) and active (microwave and radar) sensors on orbital satellites (Land et al., 2015). This opens up avenues for exploitation of LR models developed from in situ data to enable synoptic-scale monitoring of TA

25   variability and other carbonate system parameters. While such approaches have been successfully trialled for open oceans (Lee et al., 2006; Millero et al., 1998), less effort has been invested on its application at the coastal scale. The success of this application will depend largely on the resolution of the satellite data that the algorithm is applied to, the accuracy of the algorithm itself and the ability to quantify associated errors, increasing the need for high quality *in situ* measurements. Satellite observations are vulnerable to inaccuracies in coastal waters due

30   to factors including cloud cover, the presence of coloured dissolved organic matter (CDOM) and suspended sediments, the presence of both marine and terrestrial aerosols, land adjacency effects, and the electromagnetic complexity of coastal signals (in both optical and radio wave spectrum) (Schalles, 2006; Land et al., 2015). Future planned sensors with higher spatial and spectral resolution may help reduce these current limitations.

35   Our analysis has been conducted to additionally explore the possibilities of applying remote sensing platforms for the monitoring of TA. We find that this is a viable pathway, in which further study can be done, however first sampling needs to be performed on an increased spatial scale, so that algorithms can be interpolated accurately. The technology does not currently exist to remotely-sense nitrate from satellites, so BM4 is not useful when considering algorithms that can be applied to Earth observation. Nonetheless, BM4 can be utilised

40   with data from autonomous platforms equipped with nitrate sensors, such as gliders and biogeochemical

profiling floats, thus was regarded important to include in this work. These autonomous platforms provide a cost effective solution for the monitoring of TA over long time periods (with intermittent model validation), and an avenue to which the spatial distribution of TA in Australian coastal waters can be resolved.

5    The chemistry of the ocean is dynamic and varies between seasons and years, as well as through direct uptake of anthropogenic $CO_2$ emissions, and the influence of changing water temperature and salinity from climate forcing. Empirically-parameterised algorithms for TA may therefore require regular retuning to remain robust through time. The presence of ocean acidification will change TA through increasing carbonate dissolution over time (Cross et al., 2013), a process which cannot be estimated from any of the proxy variables explored in this
10   paper. This might change the required algorithm inputs significantly and increase uncertainties in algorithms over time. As such, on-going *in situ* monitoring for alkalinity and other carbonate system parameters will continue to be required to support synoptic scale approaches to monitoring the progression of ocean acidification.

**5 Conclusion**

In addressing the two main applications of the results of this paper, we have defined two different minimum sets of variables for the prediction of TA in coastal waters: S, T, and log[CHL] for applications to satellite Earth observations, and S, T, and log[N] for *in situ* applications. We have shown that a uni-parameter model is not the best model for predicting TA from ocean observations. The use of T as a predictor will improve the model significantly and the addition of a third predictor offers further improvement. We find that the influence of biological responses on the distribution of TA can be significant at some locations in Australian coastal waters, and must be considered when estimating TA. Finally, we offer recommendations for the development of robust algorithms within the locality of the 9 NRS, and present cluster models of NRS that can be used to estimate TA at a synoptic scale. These recommendations have been made by considering the results from a number of statistical parameters to assess model robustness. With this information and the models presented in this paper, more informed decisions can be made about modelling TA in Australian and other coastal waters, assisting efforts to track the progress of ocean acidification.

[revised manuscript text omitted]

**Tables**

**Table 1: Latitude and mean distributions of parameters at each NRS. Means are presented for each variable with associated standard deviations**

| NRS | Lattitude | TA | S | T | CHL | N |
|---|---|---|---|---|---|---|
| **Darwin** | -12.4 | 2265 (40) | 34.07(0.68) | 28.78(2.53) | 0.688(0.687) | 0.590(0.348) |
| **Esperance** | -33.9333 | 2337 (11) | 35.62(0.14) | 18.26(1.69) | 0.267(0.058) | 0.378(0.155) |
| **Kangaroo Island** | -35.8322 | 2355(11) | 35.84(0.19) | 16.78(1.57) | 1.083(1.561) | 0.324(0.197) |
| **Maria Island** | -42.5967 | 2324 (7) | 35.31(0.15) | 14.49(2.02) | 2.369(1.316) | 0.666(0.492) |
| **Ningaloo** | -21.99 | 2281(8) | 34.80(0.13) | 25.77(2.41) | 0.514(0.449) | 0.380(0.141) |
| **North Stradbroke Island** | -27.345 | 2324(12) | 35.48(0.21) | 22.56(2.45) | 2.917(2.432) | 0.263(0.313) |
| **Port Hacking Bay** | -34.1192 | 2326(9) | 35.47(0.13) | 18.92(2.02) | 2.439(2.271) | 0.688(0.408) |
| **Rottnest Island** | -32 | 2327(14) | 35.49(0.22) | 20.73(1.50) | 0.514(0.328) | 0.306(0.125) |
| **Yongala** | -19.3085 | 2296(32) | 35.19(0.60) | 25.86(2.37) | 0.308(0.206) | 0.248(0.145) |

**Table 2: Results for the K-S tests for differences in distribution.**
**✓ = statistically similar to observations**
**✗ = statistically different to observations**
**\* algorithms developed with low numbers of observations employed (n < 30xnumber of explanatory variables)**

| NRS | Base Model 1 | | Base Model 2 | | Base Model 3 | | Base Model 4 | | Lee et al. (2006) |
|---|---|---|---|---|---|---|---|---|---|
| | Regional | General | Regional | General | Regional | General | Regional | General | |
| Darwin | ✓ | ✗ | ✓ | ✗ | ✓* | ✗ | ✓* | ✗ | ✗ |
| Esperance | ✓ | ✗ | ✓* | ✓ | ✓* | ✓ | ✓* | ✓ | ✗ |
| Kangaroo Island | ✓ | ✗ | ✓ | ✗ | ✓* | ✗ | ✓* | ✗ | ✓ |
| Maria Island | ✗ | ✗ | ✓ | ✓ | ✓ | ✓ | ✓ | ✓ | ✗ |
| Ningaloo | ✓ | ✗ | ✓* | ✓ | ✓* | ✓ | ✓* | ✓ | ✓ |
| North Stradbroke Island | ✗ | ✗ | ✓ | ✗ | ✓ | ✗ | ✓ | ✓ | ✗ |
| Port  Hacking Bay | ✓ | ✗ | ✓ | ✓ | ✓* | ✓ | ✓ | ✓ | ✗ |
| Rottnest Island | ✓ | ✓ | ✓ | ✗ | ✓ | ✗ | ✓ | ✓ | ✗ |
| Yongala | ✗ | ✗ | ✗ | ✗ | ✓ | ✗ | ✓* | ✗ | ✗ |

**Table 3: Author recommendations for the modelling of TA in the locality of the nine NRS. Recommendations are based on a critical analysis of models using a number of statistical results, as reasoned in the text. Results are presented alongside minimum model predictions (from AIC) and the maximum effect on MAE, when BM2 is instead employed.**

| NRS | Scale | Recommended BM | Max effect on MAE using BM2 | Reasoning |
|---|---|---|---|---|
| **Darwin** | Regional | BM3 | 4.08 | Minimum model with agreeing RSE and MAE |
| **Esperance** | Regional | BM4 | -0.23 | Minimum model with agreeing RSE and MAE |
| **Kangaroo Island** | Regional | BM4 | 0.80 | Minimum model with agreeing RSE and MAE |
| **Maria Island** | Regional | BM4 | 0.32 | RSE and MAE contradict minimum model and indicate that BM4 is the best model |
| **Ningaloo** | Regional | BM4 | 0.06 | Minimum model with agreeing RSE and MAE |
| **North Stradbroke Island** | Regional | BM3 | 0.27 | Minimum model with agreeing RSE and MAE |
| **Port Hacking Bay** | Regional | BM3 | 0.04 | Minimum model with agreeing RSE and MAE |
| **Rottnest Island** | Regional | BM3 | -0.02 | RSE and MAE contradict minimum model and indicate that BM4 is the best model, although the use of BM2-4 is comparable. |
| **Yongala** | Regional | BM2 | NA | RSE and MAE contradict minimum model and indicate that BM2 is the best model. |
| **C1** | Synoptic | BM2 | NA | RSE and MAE contradict minimum model and indicate that BM2 is the best model. This makes sense as the three members have different minimum models but are all successfully modeled by BM2 at a regional scale. |
| **C2** | Synoptic | BM2 | NA | The Esperance NRS displays a different minimum model, so it is not advised to use this cluster at with BM3. |

---

## Author Comment (AC2) · 9 Oct 2017

**Supplementary information**

**Table S1: Sampling efforts for the 9 NRS used in this study. The Esperance and Ningaloo NRS were discontinued in 2013**

| NRS | Study time |
| --- | --- |
| Darwin | Jun 2011 – Feb 2016 |
| Esperance | May 2011 – Jul 2013 |
| Kangaroo Island | Oct 2008 – Sep 2016 |
| Maria Island | Apr 2009 – Sep 2016 |
| Ningaloo | Nov 2010 – Aug 2013 |
| North Stradbroke Island | Sep 2009 – Sep 2016 |
| Port Hacking Bay | Feb 2009 – Jul 2016 |
| Rottnest Island | Oct 2009 – Jul 2016 |
| Yongala | Sep 2009 – Aug 2016 |

**Table S2: Parameters and statistics for Base Model 1: Modelling TA with S**
**\* low number of observations (based on 30x[number of explanatory variables])**
**\*\* one outlier from the Esperance NRS was removed for this result**

| NRS | R | a | b | n | RSE | MAE | AIC |
|---|---|---|---|---|---|---|---|
| **General** | 0.93 | 511.61 | 51.13 | 1851 | 10.71 | 7.86 | 14035.6 |
| **Darwin** | 0.95 | 348.78 | 56.26 | 149 | 12.88 | 9.11 | 1188.5 |
| **Esperance** | 0.86 | 50.57 | 64.18 | 60 | 5.43 | 3.09\*\* | 377.3 |
| **Kangaroo Island** | 0.88 | 617.75 | 48.47 | 150 | 5.04 | 3.676 | 915.1 |
| **Maria Island** | 0.84 | 902.65 | 40.25 | 305 | 3.91 | 2.77 | 1700.5 |
| **Ningaloo** | 0.69 | 686.21 | 45.83 | 40 | 6.10 | 4.58 | 262.2 |
| **North Stradbroke Island** | 0.91 | 427.94 | 53.42 | 312 | 5.08 | 3.11 | 1903.5 |
| **Port Hacking Bay** | 0.95 | 157.34 | 61.15 | 231 | 2.72 | 2.13 | 1121.0 |
| **Rottnest Island** | 0.93 | 340.07 | 55.99 | 277 | 4.81 | 3.47 | 1660.1 |
| **Yongala** | 0.97 | 465.90 | 52.00 | 327 | 7.94 | 5.96 | 2287.3 |
| **C1** | 0.93 | 257.94 | 58.31 | 568 | 4.221 | 2.87\*\* | 3251.9 |

**Table S3: Parameters for Base Model 2: Modelling TA with S and T**
\* low number of observations (based on 30x[number of explanatory variables])
\*\* one outlier from the Esperance NRS was removed for this result

| NRS | R | a | b | c | n | RSE | MAE | AIC |
|---|---|---|---|---|---|---|---|---|
| General | 0.95 | 694.33 | 46.69 | -1.25 | 1391 | 9.00 | 5.94 | 10066.2 |
| Darwin | 0.93 | 525.48 | 52.40 | -1.6 | 87 | 13.6 | 9.85 | 706.0 |
| Esperance | 0.88 | 2.52 | 66.00 | -0.93 | 46* | 4.96 | 2.94** | 282.0 |
| Kangaroo Island | 0.89 | 636.76 | 47.74 | 0.42 | 115 | 5.11 | 3.75 | 706.4 |
| Maria Island | 0.86 | 627.74 | 48.40 | -0.89 | 229 | 3.56 | 2.47 | 1236.0 |
| Ningaloo | 0.93 | 217.92 | 60.55 | -1.67 | 25* | 3.38 | 2.61 | 136.6 |
| North Stradbroke Island | 0.90 | 613.47 | 48.20 | -0.79 | 263 | 4.82 | 2.96 | 1578.5 |
| Port Hacking Bay | 0.95 | 185.41 | 60.60 | -0.46 | 158 | 2.56 | 1.90 | 750.6 |
| Rottnest Island | 0.94 | 325.62 | 56.85 | -0.77 | 191 | 4.71 | 3.39 | 1138.8 |
| Yongala | 0.97 | 454.07 | 52.14 | 0.27 | 277 | 7.78 | 5.77 | 1927.7 |
| C1 | 0.94 | 267.8 | 58.3 | -0.56 | 395 | 4.0 | 2.74** | 2229.18 |
| C2 | 0.92 | 418.37 | 54.2 | -0.76 | 658 | 4.50 | 2.84** | 3851.22 |

**Table S4: Parameters for Base Model 3: Modelling TA with Sal, T and log(CHL)**
**\* low number of observations (based on 30x[number of explanatory variables])**
**\*\* one outlier from the Esperance NRS was removed for this result**

| NRS | R | a | b | c | d | n | RSE | MAE | AIC |
|---|---|---|---|---|---|---|---|---|---|
| General | 0.95 | 631.4 | 48.39 | -1.01 | 2.63 | 734 | 9.51 | 6.67 | 5395.1 |
| Darwin | 0.97 | 493.93 | 52.7 | -0.83 | -0.15 | 53* | 7.97 | 5.77 | 376.2 |
| Esperance | 0.87 | 59.61 | 64.42 | -0.94 | 0.59 | 37* | 5.48 | 3.18** | 236.6 |
| Kangaroo Island | 0.92 | 822.44 | 42.36 | 0.91 | 1.79 | 41* | 5.00 | 3.54 | 935.9 |
| Maria Island | 0.89 | 391.85 | 55.27 | -1.32 | 1.42 | 109 | 3.44 | 2.51 | 584.3 |
| Ningaloo | 0.92 | 263.74 | 59.09 | -1.48 | -0.30 | 20* | 3.71 | 2.64 | 114.7 |
| North Stradbroke Island | 0.94 | 458.75 | 53.12 | -0.82 | 0.47 | 91 | 3.77 | 2.69 | 505.6 |
| Port Hacking Bay | 0.95 | 148.53 | 61.57 | -0.30 | 0.31 | 73* | 2.61 | 1.86 | 353.3 |
| Rottnest Island | 0.92 | 194.57 | 60.64 | -0.82 | 1.94 | 130 | 4.90 | 3.42 | 788.2 |
| Yongala | 0.97 | 432.95 | 52.17 | 0.88 | -3.03 | 180 | 8.60 | 6.20 | 1291.6 |
| C1 | 0.93 | 204.38 | 60.14 | -0.54 | 0.41 | 395 | 4.41 | 2.88 ** | 1399.4 |
| C2 | 0.93 | 266.38 | 58.44 | -0.64 | 0.57 | 331 | 4.26 | 2.87 ** | 1905.2 |

**Table S5: Parameters for Base Model 4: Modelling TA with Sal, T and log(N)**
**\* low number of observations (based on 30x[number of explanatory variables])**
**\*\* one outlier from the Esperance NRS was removed for this result**

| NRS | R | a | b | c | d | n | RSE | MAE | AIC |
|---|---|---|---|---|---|---|---|---|---|
| General | 0.96 | 671.40 | 47.14 | -0.86 | 0.33 | 826 | 8.20 | 5.20 | 5840.3 |
| Darwin | 0.93 | 417.00 | 59.52 | -2.33 | 7.12 | 55* | 15.06 | 11.78 | 460.2 |
| Esperance | 0.90 | 33.57 | 65.60 | -1.07 | 6.36 | 37* | 5.15 | 2.68** | 232.1 |
| Kangaroo Island | 0.94 | 451.09 | 53.02 | 0.24 | 0.46 | 60* | 3.81 | 2.95 | 336.6 |
| Maria Island | 0.89 | 530.27 | 51.26 | -1.14 | -0.47 | 195 | 3.23 | 2.15 | 1016.9 |
| Ningaloo | 0.95 | 382.73 | 55.68 | -1.40 | 2.12 | 18* | 3.422 | 2.55 | 100.84 |
| North Stradbroke Island | 0.80 | 1018.88 | 37.21 | -0.68 | 0.95 | 130 | 5.24 | 3.24 | 817.7 |
| Port Hacking Bay | 0.93 | 161.94 | 61.41 | -0.74 | -0.51 | 134 | 2.59 | 1.89 | 641.27 |
| Rottnest Island | 0.93 | 344.91 | 56.27 | -0.72 | -0.00 | 141 | 4.99 | 3.54 | 859.2 |
| Yongala | 0.98 | 587.67 | 48.83 | -0.21 | 2.68 | 56* | 9.02 | 6.80 | 411.1 |
| C3 | 0.97 | 371.2 | 45.0 | -1.2 | -0.06 | 250 | 2.37 | 2.97** | 1372.0 |

**Table S6: Relative probabilities of minimising information loss for all four base models**

| NRS | BM1 | BM2 | BM3 | BM4 |
|---|---|---|---|---|
| General | 0.00 | 0.00 | 1.00 | 0.00 |
| Darwin | 0.00 | 0.00 | 1.00 | 0.00 |
| Esperance | 0.00 | 0.00 | 0.11 | 1.00 |
| Kangaroo Island | 0.00 | 0.00 | 0.00 | 1.00 |
| Maria Island | 0.00 | 0.00 | 1.00 | 0.00 |
| Ningaloo | 0.00 | 0.00 | 0.00 | 1.00 |
| North Stradbroke Island | 0.00 | 0.00 | 1.00 | 0.00 |
| Port Hacking Bay | 0.00 | 0.00 | 1.00 | 0.00 |
| Rottnest Island | 0.00 | 0.00 | 1.00 | 0.00 |
| Yongala | 0.00 | 0.00 | 0.00 | 1.00 |

**Table S7: p-values for KS tests. KS test is a two-sided test so if pvalue < 0.025 then null hypothesis is rejected then significantly different result at the 95% confidence level.**

| NRS | BM1 | | BM2 | | BM3 | | BM4 | | Lee et al. (2006) |
|---|---|---|---|---|---|---|---|---|---|
| | Regional | General | Regional | General | Regional | General | Regional | General | |
| **Darwin** | 0.773 | 0.000 | 0.435 | 0.009 | 0.164 | 0.000 | 0.198 | 0.015 | 0.003 |
| **Esperance** | 0.167 | 0.01 | 0.223 | 0.120 | 0.248 | 0.158 | 0.476 | 0.066 | 0.019 |
| **Kangaroo Island** | 0.579 | 0.000 | 0.644 | 0.000 | 1.000 | 0.001 | 0.343 | 0.000 | 0.870 |
| **Maria Island** | 0.035 | 0.000 | 0.226 | 0.071 | 0.982 | 0.258 | 0.062 | 0.162 | 0.00 |
| **Ningaloo** | 0.895 | 0.000 | 0.877 | 0.063 | 0.977 | 0.041 | 0.712 | 0.055 | 0.266 |
| **North Stradbroke Island** | 0.001 | 0.018 | 0.119 | 0.011 | 0.888 | 0.000 | 0.130 | 0.087 | 0.000 |
| **Port Hacking Bay** | 0.822 | 0.001 | 0.810 | 0.364 | 0.978 | 0.880 | 0.980 | 0.169 | 0.000 |
| **Rottnest Island** | 0.635 | 0.084 | 0.669 | 0.015 | 0.874 | 0.013 | 0.558 | 0.091 | 0.000 |
| **Yongala** | 0.000 | 0.000 | 0.002 | 0.000 | 0.147 | 0.000 | 0.440 | 0.006 | 0.000 |

**Table S8:** The p-value matrices developed in Section 2.4 of the manuscript that were used to construct Figure X. Shaded entries were replaced with a 1, whilst all other entries were forced to 0.

**M_BM1**

| | | | | | | | | |
|---|---|---|---|---|---|---|---|---|
| 0 | 0 | 0.187 | 0.011 | 0 | 0 | 0 | 0 | 0 |
| 0 | 0 | 0 | 0.026 | 0 | 0 | 0.304 | 0.165 | 0 |
| 0 | 0.003 | 0 | 0 | 0 | 0 | 0 | 0 | 0 |
| 0 | 0 | 0 | 0 | 0 | 0 | 0 | 0 | 0 |
| 0 | 0.145 | 0 | 0 | 0 | 0.001 | 0.027 | 0 | 0.002 |
| 0 | 0 | 0 | 0 | 0 | 0 | 0 | 0 | 0 |
| 0 | 0.123 | 0 | 0 | 0 | 0 | 0 | 0.265 | 0 |
| 0 | 0.906 | 0 | 0 | 0 | 0 | 0.826 | 0 | 0 |
| 0 | 0 | 0 | 0 | 0.002 | 0 | 0 | 0 | 0 |

**M_BM2**

| | | | | | | | | |
|---|---|---|---|---|---|---|---|---|
| 0 | 0 | 0.06 | 0.021 | 0 | 0.012 | 0 | 0.002 | 0 |
| 0 | 0 | 0 | 0.124 | 0 | 0.126 | 0.445 | 0.126 | 0 |
| 0 | 0.247 | 0 | 0 | 0 | 0 | 0 | 0 | 0 |
| 0 | 0 | 0 | 0 | 0 | 0 | 0 | 0 | 0 |
| 0 | 0.011 | 0 | 0.002 | 0 | 0.016 | 0.864 | 0.885 | 0.057 |
| 0 | 0.941 | 0 | 0.402 | 0 | 0 | 0.049 | 0.073 | 0 |
| 0 | 0.808 | 0 | 0.065 | 0 | 0.062 | 0 | 0.254 | 0 |
| 0 | 0.252 | 0 | 0.065 | 0.018 | 0.024 | 0.833 | 0 | 0 |
| 0 | 0 | 0 | 0 | 0 | 0 | 0 | 0 | 0 |

**M_BM3**

| | | | | | | | | |
|---|---|---|---|---|---|---|---|---|
| 0 | 0 | 0 | 0 | 0 | 0.002 | 0 | 0 | 0 |
| 0 | 0 | 0 | 0.337 | 0.008 | 0.299 | 0.67 | 0.11 | 0 |
| 0 | 0.565 | 0 | 0.09 | 0.16 | 0.012 | 0.26 | 0.548 | 0 |
| 0 | 0 | 0 | 0 | 0 | 0.321 | 0 | 0.395 | 0 |
| 0 | 0.061 | 0 | 0.52 | 0 | 0.491 | 0.963 | 0.96 | 0.069 |
| 0 | 0.945 | 0 | 0 | 0 | 0 | 0.073 | 0.984 | 0 |
| 0 | 0.988 | 0 | 0.993 | 0 | 0.958 | 0 | 0.121 | 0 |
| 0 | 0.266 | 0 | 0.022 | 0.05 | 0.449 | 0.944 | 0 | 0 |
| 0 | 0.01 | 0 | 0.005 | 0 | 0 | 0 | 0 | 0 |

**M_BM4**

| | | | | | | | | |
|---|---|---|---|---|---|---|---|---|
| 0 | 0 | 0.369 | 0 | 0 | 0.018 | 0 | 0.001 | 0.001 |
| 0 | 0 | 0.673 | 0.424 | 0.431 | 0 | 0.426 | 0.411 | 0 |
| 0 | 0.007 | 0 | 0 | 0.182 | 0 | 0.114 | 0.014 | 0 |
| 0 | 0.041 | 0.608 | 0 | 0.749 | 0 | 0 | 0 | 0 |
| 0 | 0.986 | 0.001 | 0.395 | 0 | 0.004 | 0.759 | 0.75 | 0.429 |
| 0 | 0 | 0 | 0.11 | 0.72 | 0 | 0.113 | 0.113 | 0 |
| 0 | 0 | 0 | 0.257 | 0.01 | 0.013 | 0 | 0.19 | 0 |
| 0 | 0.075 | 0.005 | 0 | 0.009 | 0 | 0 | 0 | 0 |
| 0 | 0.024 | 0 | 0.003 | 0 | 0.01 | 0.06 | 0.14 | 0 |

---

## Author Comment (AC3) · 9 Oct 2017

Author response to C. Hunt review

"This manuscript uses in-situ total alkalinity and physical/biogeochemical measurements (temperature, salinity, chlorophyll fluorescence, and nitrogen) to both test a global equation for alkalinity retrievals (Lee et al. 2006) and to develop localized equations for nine coastal sites. I think this topic is potentially of significant interest to others, and I think the authors have a very nice data set to exploit. However, I think the analysis is somewhat superficial, and I finished the paper wanted a lot more analysis and discussion that what is provided. I will address these points below, but I encourage the authors to further expand their work. The paper is generally well-written, but I list some specific comments which may improve readability."

Response: The authors thank the reviewer for expressing their concerns. This has led to major revisions, which can be viewed in the attached draft manuscript. This is the best way in which we can express such major changes.

Major Comments

"My major reservation about this work is the depth of analysis. This mostly relates to the results from the nine study sites. As noted in the manuscript, these sites are scattered along a very long coastline, distributed across a wide range of latitudes, and presumably represent contrasting conditions from the interactions of offshore ocean water and unique terrestrial and estuarine inputs and transformations. However, except for the discussion of the Yongala site (where the regression results were weakest), little to no information is presented to describe how these sites differ. The one citation referencing the sites (Page 5 Line 5, "Lynch, Morello et al. 2014) appears to be missing from the References. The reason I am left wanting more information about the physical settings is that a number of the regression coefficients are quite similar. By eye, it seems that under Base Model 2 the sites Kangaroo Island, Maria Island, and North Stradbroke Island have nearly the same regression line: is this true? If so, is this coincidence, or are there commonalities between these sites that might explain their similar results? Interestingly, the Ningaloo and Port Hacking Bay Base Model 2 results seem similar to each other, and they are on opposite sides of the continent! Considering Base Model 4, one might group Kangaroo Island, Ningaloo, and Rottnest Island, which is a much different cohort. Is there a statistical way to cluster the sites together to look for spatial trends?"

Response: Thank you for this suggestion. We have undertaken a cluster analysis of the sites to show how they relate and provide a detailed discussion in the revised manuscript. As a result the below figure and recommendations have been incorporated. Please note that a new statistic, mean absolute error (MAE) has been worked into the text. Also the missing reference has been added.

[Figure]

**Figure 1: Results from K-S tests, as described in Section 2.5. Links symbolise that the TA distributions at a particular NRS can be modelled by regressions trained from connected NRS, to significantly similar distributions. The shape of the node represents the minimum model of each NRS; circles indicate BM3 is the minimum model, while squared indicate BM4 is the minimum model.**

**Table 3: Author recommendations for the modelling of TA in the locality of the nine NRS. Recommendations are based on a critical analysis of models using a number of statistical results, as reasoned in the text. Results are presented alongside minimum model predictions (from AIC) and the maximum effect on MAE, when BM2 is instead employed.**

| NRS | Scale | Recommended BM | Max effect on MAE using BM2 | Reasoning |
|---|---|---|---|---|
| **Darwin** | Regional | BM3 | 4.08 | Minimum model with agreeing RSE and MAE |
| **Esperance** | Regional | BM4 | -0.23 | Minimum model with agreeing RSE and MAE |
| **Kangaroo Island** | Regional | BM4 | 0.80 | Minimum model with agreeing RSE and MAE |
| **Maria Island** | Regional | BM4 | 0.32 | RSE and MAE contradict minimum model and indicate that BM4 is the best model |
| **Ningaloo** | Regional | BM4 | 0.06 | Minimum model with agreeing RSE and MAE |
| **North Stradbroke Island** | Regional | BM3 | 0.27 | Minimum model with agreeing RSE and MAE |
| **Port Hacking Bay** | Regional | BM3 | 0.04 | Minimum model with agreeing RSE and MAE |
| **Rottnest Island** | Regional | BM3 | -0.02 | RSE and MAE contradict minimum model and indicate that BM4 is the best model, although the use of BM2-4 is comparable. |
| **Yongala** | Regional | BM2 | NA | RSE and MAE contradict minimum model and indicate that BM2 is the best model. |
| **C1** | Synoptic | BM2 | NA | RSE and MAE contradict minimum model and indicate that BM2 is the best model. This makes sense as the three members have different minimum models but are all successfully modeled by BM2 at a regional scale. |
| **C2** | Synoptic | BM2 | NA | The Esperance NRS displays a different minimum model, so it is not advised to use this cluster at with BM3. |

"The authors describe the site-by-site results as 'regional', but how far might that region extend around each site? How can these results be applied to locations between the study sites?"

Response: The sampling effort is not designed to answer these questions, so interpolation cannot be performed with confidence. This point is discussed in the discussion. Two interesting studies of particular interest to this question characterise a footprint of the NRS monitoring system on different time scales for different physical variables (Oke and Sakov 2012, Jones et al. 2015). However, as discussed, the role of non-conservative processes which is largely influenced by the benthos and cannot be traced along the distribution of physical parameters. Thus, extrapolation between NRS sites cannot be done with confidence, but these studies will be useful if Australian coastal waters can be divided into zones based on the non-conservative influences of each region.

"One recent paper by Carter et al. discusses a method for interpolating alkalinity data between station which may be helpful. Again, some understanding of what makes the study sites alike or different would help me understand how applicable these equations may be in other places. I also think the authors should read the paper by Alin et al. (2012), which may provide more insight for this work. Those authors also used multiple linear regression techniques to model alkalinity (and other carbonate system parameters) from physical/biogeochemical data at coastal sites."

Response: The Alin et al. 2012 paper methods cannot be applied to this work, as the spatial resolution of their data is much higher in their region of study. The study area of the Alin et al. (2012) paper corresponds to the footprint scale of one of the NRS stations. The distance in coastline by which they interpolate is not comparable to the region which we would be interpolating in our study. The Alin et al paper has enough spatial resolution to discuss spatial heterogeneity fully and has concluded that it is larger than the sampling resolution so interpolation is valid. Here, we have not been able to reach this conclusion, as the spatial heterogeneity is smaller than the sampling resolution.

**Specific Comments**
*"P2L9-10 the word "threatens/threatened" is repeated in one sentence."*
Response: Changed "Ocean acidification threatens calcifying marine organisms by hindering calcification rates, weakening the structural integrity of coral reefs and other ecosystems"

*"P2L12 what are the synoptic scales of interest?"*
Response: We have edited the text to include a definition: "The synoptic scales of interest is any scale that includes more than the locality of one NRS"

*"P3L2- Define CO2 (and format the subscript)"*
*Response: Changed*

*"P3L21-22- the phrase "contribute* [to] *a significant [presence of] calcifying fauna" seems pretty awkward"*
*Response: Changed*

"The World Heritage-listed Ningaloo Reef system and remote reef systems of the Kimberley and Pilbara coasts in Western Australia are other examples of Australia's vulnerable coral habitats. Elsewhere, sponges, bryozoans, molluscs and crustaceans contribute to a significant presence of vulnerable calcifying fauna, including some commercially significant species of abalone and scallop."

*"P3L26- This line is also pretty awkward"*
Response: Changed
*"Understanding and quantifying distributions of total alkalinity (TA), the proton deficit of seawater relative to neutrality, is an indication of how much carbon dioxide seawater can hold. Waters with higher TA are less prone to rapid change in ocean pH, as they have a higher proton deficit to "consume" the protons generated from $CO_2$ uptake, potentially offering refuge for marine biodiversity in the face of OA."*

*"P3L29-30- "TA is conservatively related to salinity"- this is an overstatement"*

Response: Changed to "Salinity is a conservative tracer within a water mass, meaning that it only experiences changes due to mixing of different water masses or through the addition or removal of freshwater. This property is often exploited through the construction of linear relationships between salinity and TA in a region for the prediction of TA."

*"P3L36- don't forget organic matter respiration too"*
Response: Changed

*"P4L24- again, what are the synoptic scales of interest?"*
Response: Addressed

"to predict TA in Australian coastal waters at regional (within the locality of the NRS) and synoptic (algorithms that combine at least 2 NRS) scales."

*"P5L23-24- how useful is an integrated phytoplankton biomass over the entire water column, if discrete alkalinity/salinity/temperature pairs are used? Will this affect the statistics, if the same CHL value is used for multiple alkalinity samples? This seems risky."*

Response: The authors acknowledge that this is a methodological error. We do have initial analysis performed using measured CHL values rather than integrated CHL values. Please see an updated draft attached in supplementary material. This has significantly changed the results, with respect to the determination of minimum models, but not the over-all message of the paper.

*"P5 and P6: the Sections "Linear Regression (LR) Analysis","Open ocean model", and "Statistical analysis" are all numbered 2.2- shouldn't they be 2.2, 2.3, and 2.4?"*
*Response: Changed*

*"P6L3- these equations are listed with little in the way of introduction. Can the authors set them up more in the text before listing them? "*
Response: Changed, please see below"

LR is well recognised as a useful predictive tool for spatial extrapolation, particularly in comparison to neural networks which are proven to have less predictive power in extrapolation (Lefèvre et al., 2008). Given the goal of enabling predictions of TA in areas of sparse *in situ* measurements, we restricted the range of input variables to those available with broad coverage from satellite Earth observation, namely T, S, and CHL. Additionally, a fourth BM that included nitrate (N) rather than CHL was included for comparison, which can be measured *in situ* using autonomous sensors. This variable choice accommodates the conservative three end-member mixing model presented in Fig.1, in addition to testing for variability due to primary production and other non-conservative coastal processes.

General and regional models for the prediction of TA were constructed from LR analysis using the four base models (BM) shown below and the lm() function in R. General models refer to those derived from a combined dataset collected from all nine NRS. Regional models refer to those derived from data collected from singular NRS. In total, 40 models were derived (the 4 base models applied to 1 general coastal model and 9 regional models).

………(equations)

*"P7L10- the term "minimum model" is a little confusing to me. It makes me think this is the minimum set of input parameters needed to accurately estimate alkalinity. Perhaps this is a statistical term I am not familiar enough with, but it seems the minimum model is just the one with the lowest AIC numbers-correct? But it may be perfectly reasonable to still use the other regression models, depending on the input data available and the user's goals."*

Response: Correct, the minimum model is the model with the lowest AIC value. The manuscript has been revised to highlight this term more clearly. Again, you are correct in saying that it is perfectly reasonable to still use the other regression models and other parameters which assess robustness should be considered. See adaption below.

«3. The **Akaike Information Criterion (AIC)** measures the relative quality of statistical models and is particularly useful when models with different numbers of variables are being compared. In calculating AIC there is a trade-off between the goodness-of-fit and the complexity of the model, adding an extra level of analysis compared to RSE. The minimum model, the model that minimises information loss, can then be determined as the model with the lowest AIC value. Using AIC values, the **relative probability of minimising information loss (RPMIL)** for each model can also be determined which normalises differences in AIC according to the number of observations collected. This allows a more intuitive and robust method for comparing models, by determining probabilities that another model is actually the minimum model given infinite data points were collected.»

*"P8- This section is so brief, it feels somewhat like an afterthought. Have the authors considered combining this with the next section into a dual Results and Discussion section? This might result in better flow from topic to topic."*
Response: The authors have edited the manuscript for flow. There is more analysis in the results section, and we hope that it has addressed this point.

*"P9L10- again, the paper by Alin et al. (2012) and other related works undermine this argument that little coastal regression work has been done."*
Response: Perhaps this is an overstatement. The text has been edited to address this.

*"P9L14-29- much of this material seems like it should be in the Introduction or perhaps Data and Methods sections- it seems out of place here."*
Response: Changed. We have moved discussion on variable choice and transformation to the methods section.

*"P9L39- are the "decreases" described here decreasing AIC values? Unclear from the text."*
Response: Changed for clarity

*"P10L37-40. This argument seems a little shaky. Seasonality in river discharge would also result in a salinity seasonality as well, not just in alkalinity. Are the authors implying that the alkalinity concentrations in river discharge vary seasonally? If so, what is the mechanism for this? Also, the authors state that this seasonality cannot be measured by remote sensing, but isn't one of the prospects held out by this paper the potential of new remotely-sensed salinity products? Why would remotely sensed salinity not show this seasonality?"*

Response: As mentioned in the text, the residuals at this station do coincide with a freshening in salinity, which is what lead us to the conclusion that the river input was the cause of error. At the Yongala NRS, there are mixing processes changing TA that are being explained by salinity, this produces the TA-S linear mixing line that has a freshwater endpoint and an oceanic endpoint. Perturbation of the system by some riverine derived variability in TA adds another direction of change to the TA, moving the intercept of the mixing line up and down but keeping the oceanic end-member constant. Thus you need an extra variable that captures this change because it is deviant from the TA-S mixing line (see below figure which has now been worked into the text). Temperature cannot be used to account for all the variability in riverine input, in this case, as the changes seen are not occurring on an annual seasonal cycle, but rather appear to be changing inter-annually. The analysis we ran shows that neither CHL nor NO3 can capture this seasonality, and it is still evident in the residuals. Upon further investigation the trends seen in the Yongala NRS residuals coincide with major flooding events, and hare highest in a period of high river flow (Logh et al. 2015) . The massive perturbations experienced by these events is a second, larger mode of variability compared to seasonal summer/winter changes in riverine input. Additionally, to answer your question, larger changes in TA would be seen in a flooding event, compared to salinity, so salinity cannot fully capture such a change. This consideration had been re-worked into the discussion.

[Figure]

**Figure 2: A depiction of a two end-member mixing model that contains an open ocean end-member (O) and a variable fresh-water end member (FW). Solid lines indicate different conservative mixing lines. Point A lies in the conservative mixing region, however an arrow indicates how it can be perturbed away from conservative mixing predictions by a non-conservative change in TA to point A'. Thus, there are three distinct modes of variability; FW variability, conservative mixing, and non-conservative changes in TA. The mixing of two oceanic end members in coastal regions it also very likely, further complicating the problem, and extending the region of variability, indicated by the addition of O' in the model, and associated mixing lines (dashed lines).**

*"P11L18- change to "in the future""*
Response: Changed

*"P11L23-38- Figure 6 might undermine several of this paper's points. For one, it only uses the general coastal model, since the authors did not attempt to interpolate their regional results over the entire domain. Also, Figure 6 shows that the actual locations of the stations where data were collected are masked out by a land mask. How sure are the authors that applying this coastal model is appropriate, given that the model data were collected in an area where remotely sensed measurements are not even available. I acknowledge that it is good to show the potential application of these relationships, but this raises the question of site selection in this study."*
Response: This figure has been removed from the manuscript

*"P13L2- Again, I am confused about the use of the term "minimum" Previously the authors state that at least temperature should be included with salinity in these equations. However, this seems to contradict that recommendation by saying that the inclusion of salinity, temperature, and CHL or N are the minimum. Also, can the authors quantify, perhaps in terms of umol alkalinity error, how much better it is to include CHL or N?"*
Response: Changed "minimum sets of variables" to "minimum models". This information can be found by comparing RSE and MAE across the two models. The results show that the robustness of each model, comparatively, it is regionally dependent. We have clarified our recommendations, as presented above, with a summarised reasoning for each conclusion, as presented in the text.

*"Figure 1- what is the inclusion of the 1000m isobath intended to show?  Why 1000m?"*
Response: The isobath was included to indicate the shape of the coastal zone and continental shelf.

*"Figure 2- the text in this figure is very small, and might not show up well in the final version"*
Response: Changed

*"Figure 3 and Figure 5- please insert a space into the µmol kg$^{-1}$ labels"*
Response: Changed

*"Figure 5- the caption says results from four models are shown, when only three are shown"*
Response: Changed

*"Table 1- I'd appreciate another table, perhaps describing the salinity, temperature, alkalinity, CHL and N data for each site.  Perhaps just some basic statistics such as range, mean, standard deviation etc."*
Response: This has been added to the manuscript. See below:

Table 1: Latitude and mean distributions of parameters at each NRS. Means are presented for each variable with associated standard deviations

| NRS | Lattitude | TA | S | T | CHL | N |
|---|---|---|---|---|---|---|
| **Darwin** | -12.4 | 2265 (40) | 34.07(0.68) | 28.78(2.53) | 0.688(0.687) | 0.590(0.348) |
| **Esperance** | -33.9333 | 2337 (11) | 35.62(0.14) | 18.26(1.69) | 0.267(0.058) | 0.378(0.155) |
| **Kangaroo Island** | -35.8322 | 2355(11) | 35.84(0.19) | 16.78(1.57) | 1.083(1.561) | 0.324(0.197) |
| **Maria Island** | -42.5967 | 2324 (7) | 35.31(0.15) | 14.49(2.02) | 2.369(1.316) | 0.666(0.492) |
| **Ningaloo** | -21.99 | 2281(8) | 34.80(0.13) | 25.77(2.41) | 0.514(0.449) | 0.380(0.141) |
| **North Stradbroke Island** | -27.345 | 2324(12) | 35.48(0.21) | 22.56(2.45) | 2.917(2.432) | 0.263(0.313) |
| **Port Hacking Bay** | -34.1192 | 2326(9) | 35.47(0.13) | 18.92(2.02) | 2.439(2.271) | 0.688(0.408) |
| **Rottnest Island** | -32 | 2327(14) | 35.49(0.22) | 20.73(1.50) | 0.514(0.328) | 0.306(0.125) |
| **Yongala** | -19.3085 | 2296(32) | 35.19(0.60) | 25.86(2.37) | 0.308(0.206) | 0.248(0.145) |

*"Figure S1- Might this be better shown in a table?  Or at least could this information be briefly described in the text?  This figure seems a bit superfluous to include."*

Response: Changed. We have presented the study period as a range in table form as suggested.
*"Table S1-S4- while these tables contain a lot of information, they are also really the heart of the paper's analysis.  Seems a little strange to bury them in the Supplementary Information."*

Response: The key message of the paper is not the regression models themselves, but rather their ability to predict TA at regional and continental scales. It was considered to include tables in the text, however it was decided that the parameters would only be considered and used by a minority of readers. Thus, AIC values and

RSE values were chosen to be better displayed in the text as these are the parameters required to formulate the results of the paper.

*"Also, do the terms Intercept, S, T etc. in these tables refer to the terms d, a, b respectively in the model equations listed on Page 6 of the main text? If so, please use consistent terms between the two."*

Response: Changed.

**References**

Jones, E.M., Doblin, M.A., Matear, R. and King, E.: Assessing and evaluating the ocean-colour footprint of a regional observing system, *J. Marine Syst.*, *143*, pp.49-61, 2015.

Lough, J.M., Lewis, S.E. and Cantin, N.E.: Freshwater impacts in the central Great Barrier Reef: 1648–2011. Coral Reefs, 34(3), 739-751, 2015.

Oke, P.R. and Sakov, P.: Assessing the footprint of a regional ocean observing system. *J. Marine Syst.*, *105*, 30-51, 2012.

---

## Author Comment (AC4) · 9 Oct 2017

"The manuscript "Estimating total alkalinity for coastal ocean acidification monitoring at regional to continental scales in Australian coastal waters" by K. Baldry and co-workers reports and discusses an approach to estimate total alkalinity (TA) in coastal waters with the intention to characterize vulnerability or resilience of such waters with respect to ocean acidification. The authors employ field data of TA and of further ocean water properties to derive a suite of models to parameterize TA. In summary I have serious concerns and reservations with the paper, such that I unfortunately cannot recommend publication of the paper at its present state. I do see some potential to

improve the paper, however this would require a major overhaul of the paper. I hope the authors could make some use of my comments in order to do so. 1: In fact the paper stops, where it should start. As far as I understood the paper, the paper only takes TA and related properties to derive a suite of algorithms/models to re(!)compute TA. The following discussion then compares the TA computation with the observed TA, but nothing goes beyond the use to the variables, which have been used to train the regressions. Thus, there is no estimation of TA, so far it appears to be a recompilation only. I was searching for some time for the application of these regressions, which goes beyond training, and eventually discovered a 2-line statement about Figure 6 – which is for illustrative purposes only? Frankly, what is the usefulness of a colorful Figure for illustrative purposes? In my view this is the point where the paper should start, including detailed validation with respect to data, including data, which have not been used to train the regressions. In essence, anything prior to figure 6 is an extended methods section."

Response: The paper aims to assess the validity of employing a uni-parameter model, relating alkalinity to salinity, for the study of the ocean carbonate system in costal areas, particularly around Australia. In no way was it suggested that the authors were characterising the biogeochemistry of the Australian coastline by constructing a distribution from empirical relationships, but rather exploring its heterogeneity and the sources of error that could arise through the use of a well accepted method due to coastal processes. This is why TA was recomputed from the algorithms, to assess if the modeled TA had a significantly different distribution to observed TA, and that variability had been captured effectively by the model. The test was bootstrapped to account for the paired data set. By employing the use of AIC values we attempt to prevent over-fitting, as the calculation for AIC has a penalty term for the addition of explanatory variables. In statistical theory, this is equivalent to minimizing the cross-validation term (CV), a common term used to choose optimum models, and the term employed by Lee et al. (2006). This is proven in Stone (1977).

Figure 6 is only intended to highlight the capabilities, if enough effort is put into understanding the empirical estimation of TA. As mentioned in the manuscript, the authors understand that unreasonable extrapolation is performed in constructing this figure, particularly as the heterogeneity around the coastline is larger than the number of points sampled, and the distance around which the algorithms hold has not been explored due to the absence of data. The authors agree that this plot can be easily misinterpreted and draw attention away from the main objectives; hence it has been removed in the revised manuscript.

We thank the reviewer for their concerns regarding the objectives of the manuscript. The review process has given us the opportunity to expand the analysis. . A new draft manuscript is attached in the author comments of the main discussion for your consideration. However, the authors respectfully feel that any further discussion on the distribution and drivers of TA in the coastal waters of Australia would be too speculative, as we have shown that the heterogeneity around the entirety of the Australian coastline cannot be fully captured with the current sampling effort.

"2: The application of the newly obtained regressions to independent data is particularly relevant to such an approach, as the causal relationship between TA and the regression properties is not clear or even not given. The extrapolation of pure empirical relationships, i.e., regression coefficients, bears the massive risk, that these only hold true within their framework of training, or trained data. It might well be the case that the extrapolation does work very well, it could also be the opposite. Figure 6 should have been the first step to open this discussion. Along these lines the justification or even explanation of regression parameters falls short, specifically with respect to the non-conservative parameters: A: Amongst the most powerful characteristics of TA is its temperature INDEPENDENCE. Open ocean TA vs. temperature relationships are not much more than masked TA-depth relationships, if at all. If you refer to temperature as (partial) TA proxy, please explain and justify, why it is used. What about seasonality, and which processes does such a relationship mimic? B: In a similar manner the used
of Chl-a as partial TA proxy should be discussed. What thought does support this? Why are water column inventories used rather than actual concentrations? The satellites do not sense the water column inventory of Chla, they "see" the upper most layer? Also an important point to be considered here are the problems of remotely sensed Chl-a values in coastal waters (case 1 vs case 2 waters). The authors mention initially that Australia's coastlines pans 33 degrees in latitude, which likely causes vastly different organic matter composition of such coastal waters. C: The use of nitrate should be justified here as well. What does it stand for, maybe as runoff proxy, or proxy for biological metabolism?"

Response: Thank you for raising this query, the authors felt that enough literature around the use of chosen variables had been published and widely accepted, and that the introduction to the manuscript explained the reasoning behind variable choice, however this is evidently not the case. We have included a section for model justification in the methods as a result. Please also see the discussion below:

The use of a uni-parameter model for the estimation of TA in coastal area is generally inadequate as usually there are more than two water masses mixing, with e.g. upwelling and river end members having variable effects over time, which alone is enough reason for the model to fail. In a region that exhibits the mixing of n end-members, a model with at least n-1 conservative explanatory variables must be employed in order to account for all conservative variability. Temperature is a conservative variable, and yes, it does not directly effect TA, but rather it indirectly is related to TA, when a cold water mass mixes with a warmer water mass, such as when upwelling occurs, which we know has seasonal variability and is not constant (Jiang et al. 2014, Lee et al. 2006, Millero et al. 1998, Ingrosso et al. 2016). Thus, the inclusion of two conservative parameters seems relevant to model the coastal region, which most often experiences the mixing of at least three water masses (river end-member, coastal end-member, offshore end-member, upwelling water mass). The inclusion of a third explanatory variable ($NO_3^-$ or Chl-a) offers a seasonal tracer to account for seasonal variations in TA

due to primary production, or river flow (river flow can only be captured by NO3- as discussed in the manuscript). To illustrate the casual relationships of the chosen variables (S, Temp, Chl-a) more clearly we will include a conceptual graphic in the introduction. As Landimayer et al. (2016) explains, indirect linkages have been exploited to study environmental surrogates in numerous cases, and is a viable method. The complexity of the problem can be illustrated by Figure 1 which is now included in the introduction. Please find the full caption below:

Figure 1: A depiction of a two end-member mixing model that contains an open ocean end-member (O) and a variable fresh-water end member (FW). Solid lines indicate different conservative mixing lines. Point A lies in the conservative mixing region, however an arrow indicates how it can be perturbed away from conservative mixing predictions by a non-conservative change in TA to point A'. Thus, there are three distinct modes of variability; FW variability, conservative mixing, and non-conservative changes in TA. The mixing of two oceanic end members in coastal regions it also very likely, further complicating the problem, and extending the region of variability, indicated by the addition of O' in the model, and associated mixing lines (dashed lines).

"Why are water column inventories used rather than actual concentrations? The satellites do not sense the water column inventory of Chla, they "see" the upper most layer?"

The authors note that the use of a water column inventory for Chla was a mistake in the previous version and such a method would not be statistically viable, as the TA observations were taken from multiple depths within this water column, so taking the inventories decreases model power. This was also bought to the attention by Reviewer 1, and consequently the algorithms presented for Base Model 3 now use discrete Chla measurements. Please see the manuscript in the author comments for changes.

References Jiang, Z. P., Tyrrell, T., Hydes, D. J., Dai, M. H., and Hartman, S. E.: Variability of alkalinity and the alkalinity-salinity relationship in the tropical and subtropical surface ocean, Glob. Biogeochem. Cycle, 28, 729-742, 2014. Lee, K., Tong, L. T.,

Millero, F. J., Sabine, C. L., Dickson, A. G., Goyet, C., Park, G. H., Wanninkhof, R., Feely, R. A., and Key, R. M.: Global relationships of total alkalinity with salinity and temperature in surface waters of the world's oceans, Geophys. Res. Let., 33, 5, 2006. Lindenmayer, D., Pierson, J., Barton, P., Beger, M., Branquinho, C., Calhoun, A., Caro, T., Greig, H., Gross, J., Heino, J. and Hunter, M.: A new framework for selecting environmental surrogates. Sci. Total Environment, 538, 1029-1038, 2015. Ingrosso, G., Giani, M., Cibic, T., Karuza, A., Kralj, M. and Del Negro, P.: Carbonate chemistry dynamics and biological processes along a river–sea gradient (Gulf of Trieste, northern Adriatic Sea), J. Marine Syst., 155, 35-49, 2016. Millero, F. J., Lee, K., and Roche, M.: Distribution of alkalinity in the surface waters of the major oceans, Mar. Chem., 60, 111-130, 1998. Stone, M.: An asymptotic equivalence of choice of model by cross-validation and Akaike's criterion, J. R. Stat. Soc. B Met., 44-47, 1977.

[Figure]

**Fig. 1.** A depiction of a two end-member mixing model that contains an open ocean end-member (O) and a variable fresh-water end member (FW). Solid lines indicate different conservative mixing lines.